# Cryo-sensitive aggregation triggers NLRP3 inflammasome assembly in cryopyrin-associated periodic syndrome

Tadayoshi Karasawa*, Takanori Komada, Naoya Yamada, Emi Aizawa, Yoshiko Mizushina, Sachiko Watanabe, Chintogtokh Baatarjav, Takayoshi Matsumura, Masafumi Takahashi*

Division of Inflammation Research, Center for Molecular Medicine, Jichi Medical University, Tochigi, Japan

**Abstract** Cryopyrin-associated periodic syndrome (CAPS) is an autoinflammatory syndrome caused by mutations of NLRP3 gene encoding cryopyrin. Familial cold autoinflammatory syndrome, the mildest form of CAPS, is characterized by cold-induced inflammation induced by the overproduction of IL-1β. However, the molecular mechanism of how mutated NLRP3 causes inflammasome activation in CAPS remains unclear. Here, we found that CAPS-associated NLRP3 mutants form cryo-sensitive aggregates that function as a scaffold for inflammasome activation. Cold exposure promoted inflammasome assembly and subsequent IL-1β release triggered by mutated NLRP3. While $K^+$ efflux was dispensable, $Ca^{2+}$ was necessary for mutated NLRP3-mediated inflammasome assembly. Notably, $Ca^{2+}$ influx was induced during mutated NLRP3-mediated inflammasome assembly. Furthermore, caspase-1 inhibition prevented $Ca^{2+}$ influx and inflammasome assembly induced by the mutated NLRP3, suggesting a feed-forward $Ca^{2+}$ influx loop triggered by mutated NLRP3. Thus, the mutated NLRP3 forms cryo-sensitive aggregates to promote inflammasome assembly distinct from canonical NLRP3 inflammasome activation.

## Editor's evaluation

Gain of function mutations in NLRP3 are associated with a group of autoinflammatory diseases called the cryopyrin-associated periodic syndromes (CAPS). Karasawa and colleagues reveal that CAPS-associated NLRP3 mutants form cryo-sensitive aggregates that promote NLRP3 inflammasome assembly and Caspase-1 activation through elegant immunofluorescence studies, providing mechanistic insights into CAPS.

*For correspondence:
tdys.karasawa@jichi.ac.jp (TK);
masafumi2@jichi.ac.jp (MT)

**Competing interest:** The authors declare that no competing interests exist.

## Introduction

The cryopyrin-associated periodic syndromes (CAPS) are a spectrum of rare diseases consisting of three clinically defined autosomal dominant disorders: familial cold autoinflammatory syndrome (FCAS), Muckle-Wells syndrome (MWS), and chronic infantile neurological, cutaneous, and articular syndrome (CINCA) (*Kuemmerle-Deschner et al., 2017*). These three syndromes can be classified according to severity. FCAS is the mildest form of CAPS and is characterized by cold-induced fever, arthralgia, urticaria, and conjunctivitis. MWS is accompanied by systemic amyloidosis and progressive hearing loss. CINCA is the most severe phenotype and is characterized by CNS inflammation, bone deformities, and chronic conjunctivitis. Genetic causes of these disorders are gain-of-function mutations in the NLRP3 gene, encoding cryopyrin (*Hoffman et al., 2001*; *Brydges et al., 2009*;

*Kuemmerle-Deschner, 2015*). The mutated NLRP3 protein causes overproduction of IL-1β, resulting in systemic inflammatory characteristics, such as recurrent fever, rash, conjunctivitis, and arthralgia.

NLRP3 forms a multi-protein molecular complex called 'NLRP3 inflammasome' (*Schroder and Tschopp, 2010*). NLRP3 inflammasome is composed of NLRP3, apoptosis-associated speck-like protein containing a caspase recruitment domain (ASC) which acts as an adaptor protein, and the cysteine proteinase caspase-1, and functions as a scaffold for caspase-1 activation (*Schroder and Tschopp, 2010*; *Karasawa and Takahashi, 2017*). The assembly of inflammasome complex promotes oligomerization and auto-processing of caspase-1. The active caspase-1 processes precursors of inflammatory cytokines IL-1β and IL-18 and converts them to their mature forms. Another critical role of caspase-1 is the processing of gasdermin D (GSDMD) (*Liu et al., 2016*; *Broz et al., 2020*). The processed amino-terminal domain of GSDMD binds to the plasma membrane and forms pores. Therefore, caspase-1-mediated GSDMD pore induces the release of cytosolic content and subsequent necrotic cell death called pyroptosis.

Although NLRP3 was initially identified as a causative gene of CAPS (*Hoffman et al., 2001*), the function of NLRP3 had been unclear because CAPS is a rare disease. In 2006, however, Tschopp and his colleagues found that monosodium urate crystals activate NLRP3 inflammasome (*Martinon et al., 2006*). Since this finding, many studies have clarified the pivotal role of NLRP3 inflammasome in inflammatory responses in both host defense and sterile inflammatory diseases. Other investigators and we have demonstrated the pathophysiological role of NLRP3 inflammasome in cardiovascular and renal diseases (*Duewell et al., 2010*; *Usui et al., 2012*; *Usui et al., 2015*; *Komada et al., 2014*; *Komada et al., 2015*).

Despite many findings regarding molecular mechanisms and the pathophysiological role of the NLRP3 inflammasome, the disease mechanisms of CAPS are not fully understood. In particular, although FCAS is characterized by cold exposure-induced recurrent fever and inflammation, the mechanisms by which exposure to cold regulates NLRP3 inflammasome in FCAS remain unclear (*Rosengren et al., 2007*; *Brydges et al., 2013*). In the present study, we have found that CAPS-associated NLRP3 mutants form cryo-sensitive aggregates, which function as scaffolds for NLRP3 inflammasome assembly. The aggregation of the mutated NLRP3 is sensitive to $Ca^{2+}$. Therefore, mutated NLRP3 triggers inflammasome assembly driven by $Ca^{2+}$ influx-mediated feed-forward regulation.

## Results

### CAPS-associated NLRP3 mutants form cryo-sensitive foci

To investigate the pathophysiological role of CAPS-associated NLRP3 mutants (*Cordero et al., 2018*), we generated cell lines expressing fusion proteins of NLRP3 mutants and a green monomeric protein, mNeonGreen (*Figure 1—figure supplement 1A and B*). We found that FCAS-associated NLRP3-L353P and -Y563N, as well as CINCA-associated NLRP3-D303N and -Y570C formed foci without any stimulation, while wild type (WT)-NLRP3 is expressed diffusely (*Figure 1A* and *Figure 1—figure supplement 1C*). On the other hand, ASC-GFP forms a single speck per cell. To analyze the localization of NLRP3 during NLRP3 inflammasome activation without being affected by ASC, we generated *ASC KO* THP-1 cells (*ASC KO/EF-1-NLRP3-mNeonGreen*-THP-1). NLRP3-D303N formed foci in *ASC KO* THP-1 cells, whereas WT-NLRP3 did not form foci upon stimulation by the NLRP3 activator nigericin, indicating that the foci are distinct from canonical inflammasome assembly (*Figure 1B and C*). To assess whether the foci formation is cryo-sensitive, the transduced cells were exposed to cold temperature (32°C) for 24 hr. A considerable number of foci were detected in CINCA-associated D303N and Y570C mutant-expressing cells under normal temperature (37°C). Although the number of foci was increased in all of the mutant-expressing cells under cold exposure, the sensitivity to cold exposure was prominent in FCAS-associated mutants (*Figure 1D–I*, and *Figure 1—figure supplement 1D-G*). The number of foci formed was weakly associated with expression levels of NLRP3 as indicated by fluorescence of mNeonGreen. In contrast, speck formation by ASC-GFP was not affected by cold exposure (*Figure 1J* and *Figure 1—figure supplement 1H*). These results suggest that CAPS-associated NLRP3 mutants form cryo-sensitive foci consistent with disease severity and characteristics.

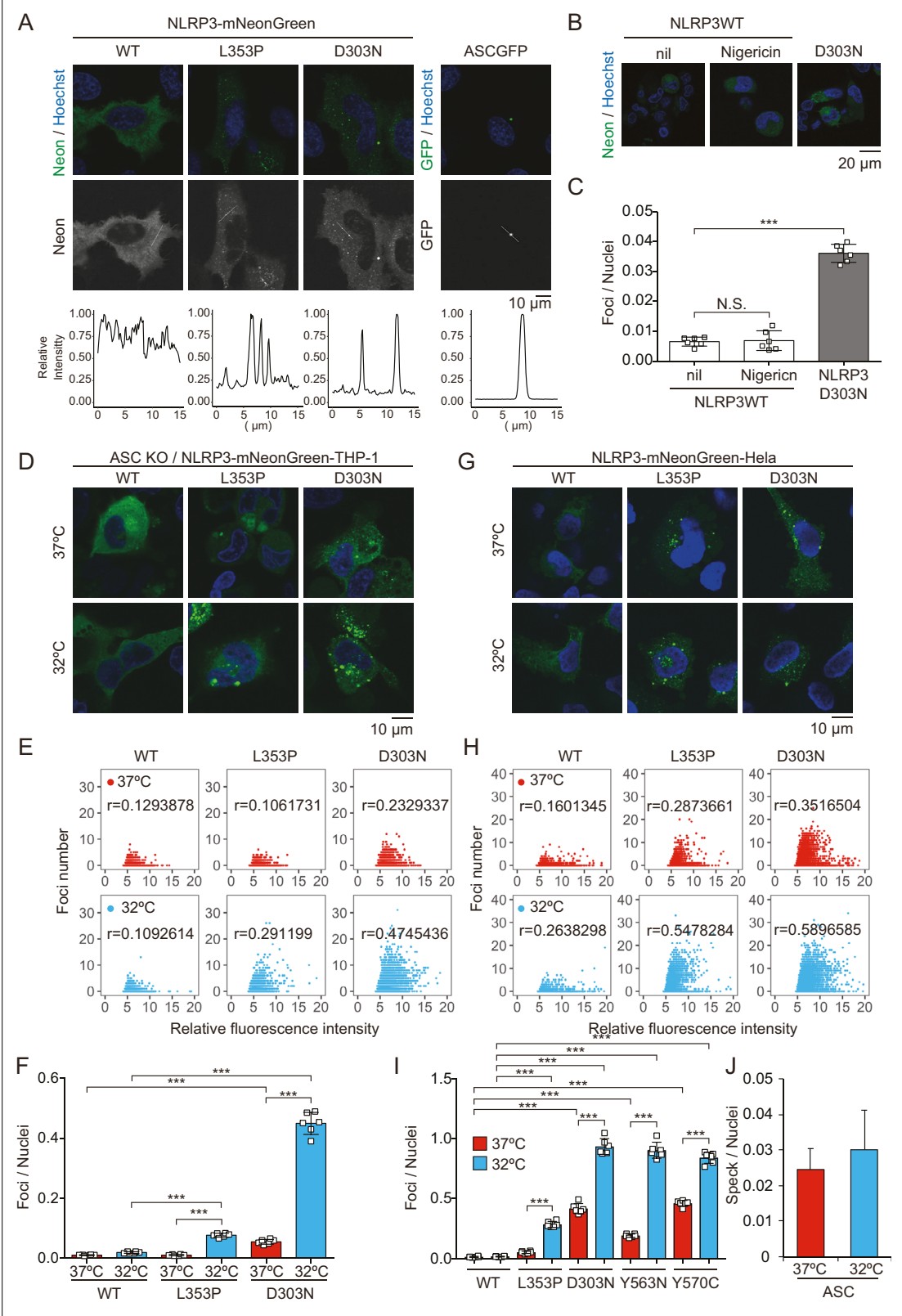

**Figure 1.** Cryopyrin-associated periodic syndrome-associated NLRP3 mutants form cryo-sensitive foci. (**A**) *EF1-NLRP3-WT-, NLRP3-L353P-,* or *NLRP3-D303N-mNeonGreen-*HeLa cells or *EF1-ASC-GFP-*HeLa cells were analyzed by confocal microscopy. Line profiles of foci or specks in the images were analyzed. (**B and C**) *ASC KO/EF1-NLRP3-WT-* or *NLRP3-D303N-mNeonGreen-*THP-1 cells were differentiated with 200 nM phorbol-12-myristate-13-acetate for 24 hr and then treated with nigericin for 6 hr. (**B**) Representative images by confocal microscopy. (**C**) The number of foci was counted by

*Figure 1 continued on next page*

*Figure 1 continued*

high-content analysis. (D–F) Differentiated *ASC KO/EF1-NLRP3-WT-, NLRP3-L353P-,* or *NLRP3-D303N-mNeonGreen*-THP-1 cells were cultured at 37 or 32°C for 24 hr. (G–I) *EF1-NLRP3-WT-, NLRP3-L353P-, NLRP3-D303N-, NLRP3-Y563N-,* or *NLRP3-Y570C-mNeonGreen*-HeLa cells were cultured at 37 or 32°C for 24 hr. (**D and G**) Representative images by confocal microscopy. (**E, F, H, and I**) The number of foci and the fluorescence intensity of the cells were analyzed by high-content analysis. Pearson correlation coefficients are shown. (**J**) *EF1-ASC-GFP*-HeLa cells were cultured at 37 or 32°C for 24 hr. The number of nuclei and speck was counted. (**C, F, I, and J**) Data are expressed as the mean ± SD. ***p<0.005 as determined by two-way ANOVA with a post hoc test. Data are representative of three independent experiments. WT, wild type.

The online version of this article includes the following source data and figure supplement(s) for figure 1:

**Source data 1.** Source data for *Figure 1E*.

**Source data 2.** Source data for *Figure 1H*.

**Source data 3.** Source data for *Figure 1J*.

**Figure supplement 1.** Expression of NLRP3-mNeonGreen and apoptosis-associated speck-like protein containing a caspase recruitment domain (ASC)-GFP.

**Figure supplement 1—source data 1.** Source data for *Figure 1—figure supplement 1B*.

## CAPS-associated NLRP3 mutants form aggregates

Since NLRP3 has a pyrin domain (PYD) scaffold domain (*Figure 2—figure supplement 1A*), a common feature of molecules that forms aggregates or liquid-liquid phase separation (LLPS), we hypothesized that CAPS-associated NLRP3 mutants form aggregates or LLPS (*Alberti et al., 2019*). To test this hypothesis, we performed fluorescence recovery after photobleaching (FRAP) analysis. After induction of foci formation by cold exposure, some of the NLRP3-L353P- and D303N-mNeonGreen-foci were bleached. The fluorescence in the bleached area was not recovered, indicating that the foci are aggregates (*Figure 2A and B*, *Figure 2—figure supplement 2A and B*). The wholly bleached area of NLRP3-L353P-mNeonGreen-foci was also not recovered (*Figure 2—figure supplement 2C and D*). Similar results are obtained from FRAP analysis of ASC speck, initially reported to be aggregates (*Masumoto et al., 1999*; *Figure 2C and D*). In contrast, diffusely expressed mNeonGreen was recovered after photobleaching (*Figure 2—figure supplement 2E and F*). In both NLRP3-L353P foci and ASC speck, the exchange of protein between bleached and unbleached area was not detected (*Figure 2E and F*). Furthermore, 1,6-hexanediol, an LLPS inhibitor, did not affect foci formation of NLRP3-mNeonGreen (*Figure 2—figure supplement 2G*). These results suggest that foci formed by CAPS-associated NLRP3 mutants are aggregates.

## Aggregates formed by CAPS-associated NLRP3 mutants are the scaffold for inflammasome activation

Next, we investigated whether aggregates formed by mutated NLRP3 function as a scaffold for inflammasome assembly and trigger subsequent IL-1β release. In order to analyze colocalization of NLRP3 and ASC, we developed THP-1 cells harboring two reporters; *TRE-NLRP3-mNeonGreen* and *EF1-ASC-BFP*. After induction of NLRP3-L353P-mNeonGreen by doxycycline (DOX), ASC speck was colocalized with the NLRP3 mutant-formed aggregate (*Figure 3A–C*). To exclude the possibility that NLRP3 mutant aggregation is due to its fluorescent tag, the cells expressing NLRP3 mutants under TET-ON promoter were developed (*Figure 3—figure supplement 1A and B*). In accordance with the cryo-sensitive formation of aggregates by NLRP3-L353P, insoluble complex formation and oligomerization of ASC induced by NLRP3-L353P were enhanced by cold exposure (*Figure 3D and E*). Further, NLRP3 mutants formed insoluble complexes in *ASC KO* THP-1 cells (*Figure 3—figure supplement 1C*). In contrast, the ASC oligomerization induced by nigericin was attenuated under cold exposure (*Figure 3F*). ASC-speck formation was further assessed by fusion protein of ASC-GFP reporter. Similarly, NLRP3-L353P-induced ASC-speck formation was increased under cold exposure (*Figure 3G*). Moreover, cold exposure enhanced IL-1β release induced by the NLRP3-L353P, whereas nigericin- and nanosilica-induced IL-1β release was restrained under cold exposure (*Figure 3H* and *Figure 3—figure supplement 1D*). Cold exposure also enhanced IL-1β release in CINCA-associated NLRP3-D303N-expressing cells (*Figure 3—figure supplement 1E*). In contrast, WT-NLRP3 failed to induce IL-1β release (*Figure 3—figure supplement 1F and G*). These results demonstrate that cryo-sensitive aggregates formed by CAPS-associated NLRP3 mutants function as a scaffold for inflammasome activation and induce subsequent IL-1β release.

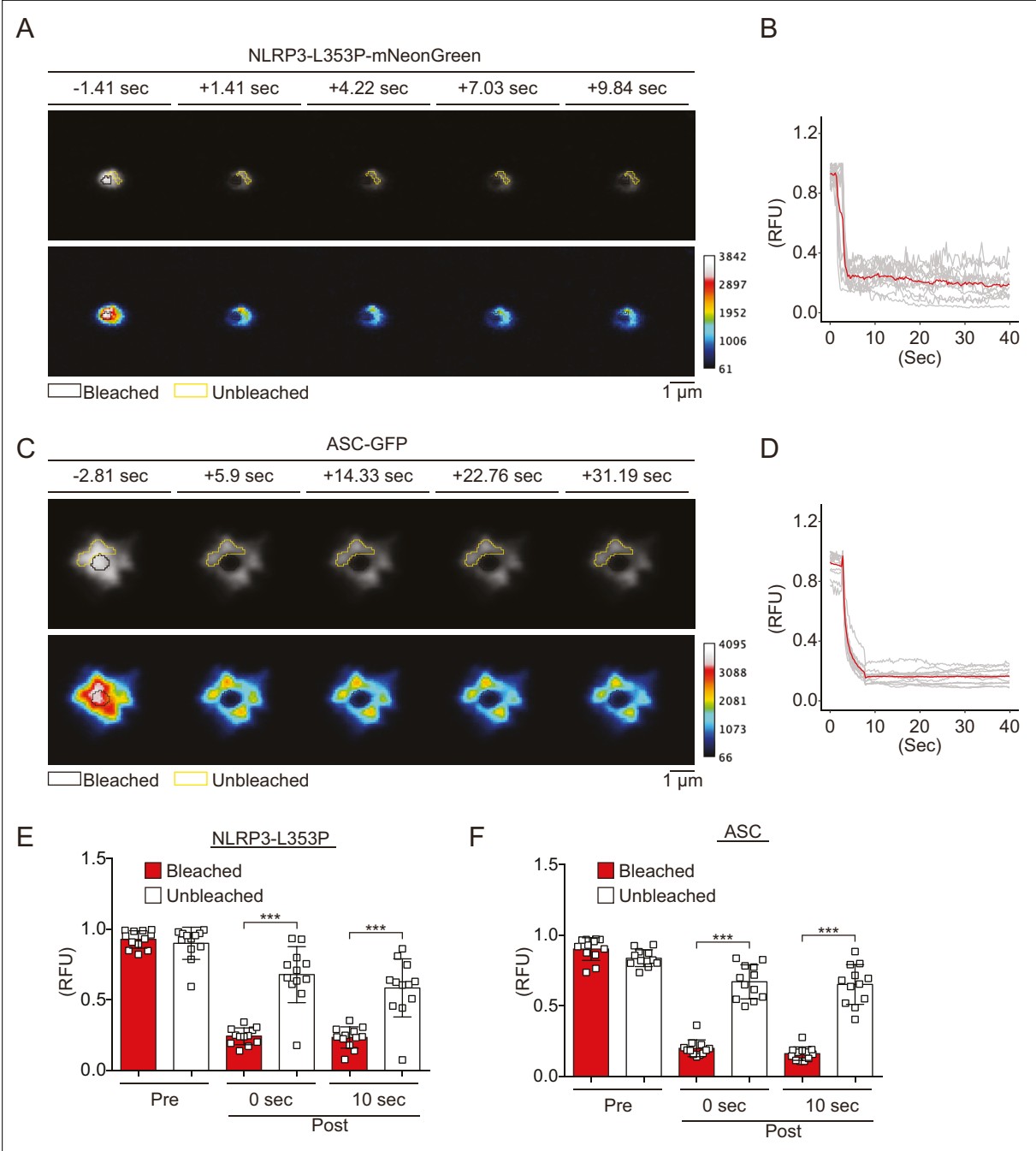

**Figure 2.** Cryopyrin-associated periodic syndrome-associated NLRP3 mutants form aggregates. (**A–F**) *EF1-NLRP3-L353P-mNeonGreen-* or *EF1-ASC-GFP*-HeLa cells were cultured at 32°C for 24 hr. Foci or specks in the cells were analyzed by fluorescence recovery after photobleaching. Representative images of (**A**) foci formed by NLRP3-L353P-mNeonGreen or (**C**) specks formed by apoptosis-associated speck-like protein containing a caspase recruitment domain (ASC)-GFP before and after photobleaching. The bleached and unbleached areas are shown in black lines and yellow lines, respectively. Plots of relative fluorescence units during photobleaching of (**B and E**) NLRP3-L353Pm-NeonGreen (n=12) and (**D and F**) ASC specks (n=12). (**B and D**) The red line represents mean values, and the gray lines represent each measurement. (**E and F**) Data are expressed as the mean ± SD. ***p<0.005 as determined by two-way ANOVA with a post hoc test. Data are from three independent live-cell imaging.

The online version of this article includes the following source data and figure supplement(s) for figure 2:

**Source data 1.** Source data for *Figure 2B*.

**Source data 2.** Source data for *Figure 2D*.

**Figure supplement 1.** Domains composing NLRP3.

*Figure 2 continued on next page*

*Figure 2 continued*

**Figure supplement 2.** Cryopyrin-associated periodic syndrome-associated NLRP3 mutants form aggregates.

**Figure supplement 2—source data 1.** Source data for *Figure 2—figure supplement 2B*.

**Figure supplement 2—source data 2.** Source data for *Figure 2—figure supplement 2F*.

## Canonical inflammasome pathway is dispensable for NLRP3 mutant-mediated inflammasome assembly

To elucidate the regulatory mechanisms of NLRP3 mutant-mediated inflammasome activation under cold exposure, we assessed the involvement of $K^+$ efflux. Unexpectedly, inhibition of $K^+$ efflux failed to prevent NLRP3-L353P-induced IL-1β release, while it inhibited nigericin-induced IL-1β release (*Figure 4—figure supplement 1A*). Since NEK7 functions as an essential component of $K^+$ efflux-mediated NLRP3 inflammasome, we developed NEK7-deficient cells (*Figure 4—figure supplement 1B-D*). However, deficiency of NEK7 failed to inhibit NLRP3 mutant-mediated IL-1β release, although it inhibited nigericin-induced IL-1β release (*Figure 4A and B*, *Figure 4—figure supplement 1E*). Recent studies have suggested that NLRP3 is localized on the membrane of trans-Golgi network (TGN) via its polybasic linker, and TGN dispersion plays a critical role in canonical inflammasome assembly (*Andreeva et al., 2021*; *Chen and Chen, 2018*). To investigate the involvement of TGN in NLRP3 mutant-mediated inflammasome activation, we developed the mutants lacking three lysine residues (K129, K131, and K132) in the polybasic linker of NLRP3 by introducing alanine (*Figure 4—figure supplement 1F*). However, both linker mutants of L353P and D303N formed aggregates (*Figure 4C*). Although the foci formation by NLRP3-D303N was slightly decreased by the polybasic linker mutation, cold exposure enhanced the formation of foci even in the polybasic linker mutants (*Figure 4D and E*). These results suggest that $K^+$ efflux-mediated canonical inflammasome pathway is dispensable for NLRP3 mutant-mediated inflammasome assembly.

## $Ca^{2+}$ is required for NLRP3 mutant-mediated inflammasome assembly

To further explore upstream pathways of NLRP3 mutant-mediated inflammasome assembly, the involvement of lysosomal destabilization, mitochondrial reactive oxygen species (ROS) generation, and $Ca^{2+}$ mobilization was investigated. Among these, EGTA, a chelator of $Ca^{2+}$, inhibited NLRP3-L353P-induced IL-1β release, while inhibition of lysosomal or mitochondrial ROS pathway did not suppress it (*Figure 5A*, *Figure 5—figure supplement 1A-C*). Moreover, decreased IL-1β release by NLRP3-L353P under $Ca^{2+}$-depleted conditions was restored by $Ca^{2+}$ supplementation at 32°C (*Figure 5B*). Next, we assessed whether deprivation or supplementation of $Ca^{2+}$ alters ASC oligomerization. $Ca^{2+}$ deprivation by EGTA attenuated NLRP3-L353P-induced ASC oligomerization, whereas reduced ASC oligomerization in $Ca^{2+}$-depleted conditions was restored by $Ca^{2+}$ supplementation (*Figure 5C* and *Figure 5—figure supplement 1D*). The requirement of $Ca^{2+}$ for inflammasome assembly was also confirmed by the use of ASC-GFP reporter cells (*Figure 5D* and *Figure 5—figure supplement 1E*). The effect of $Ca^{2+}$ on NLRP3 aggregation was analyzed using NLRP3-mNeonGreen reporter cells. Cryo-sensitive aggregation of NLRP3-L353P was decreased by $Ca^{2+}$ depletion (*Figure 5E and F*), even though cryo-sensitive aggregation of NLRP3-L353P was detected in $Ca^{2+}$-depleted conditions (*Figure 5—figure supplement 2A*). Moreover, DOX-induced NLRP3-L353P aggregation and ASC-speck formation were attenuated by $Ca^{2+}$ depletion (*Figure 5G and H*). Although a similar dependency on $Ca^{2+}$ was also detected in the CINCA-associated NLRP3-D303N (*Figure 5—figure supplement 1F* and *Figure 5—figure supplement 2B and C*), the impact of $Ca^{2+}$ depletion was less effective in IL-1β release by NLRP3-D303N. These results suggest that $Ca^{2+}$ regulates the aggregation of CAPS-associated NLRP3 mutants and subsequent activation of NLRP3 inflammasome.

## $Ca^{2+}$ influx is provoked during mutated NLRP3-mediated inflammasome assembly

To further investigate the role of $Ca^{2+}$ in mutated NLRP3-mediated inflammasome activation, we monitored changes in $Ca^{2+}$ levels using Fluo-8, a fluorescent $Ca^{2+}$ indicator. After DOX-mediated induction of NLRP3-L353P or NLRP3-D303N, $Ca^{2+}$ increase was clearly detected (*Figure 6A–C* and *Figure 6—figure supplement 1A* and B). This increased intracellular $Ca^{2+}$ is due to influx because $Ca^{2+}$ increase was not observed in the absence of extracellular $Ca^{2+}$ (*Figure 6D–F*, *Video 1*). The increased

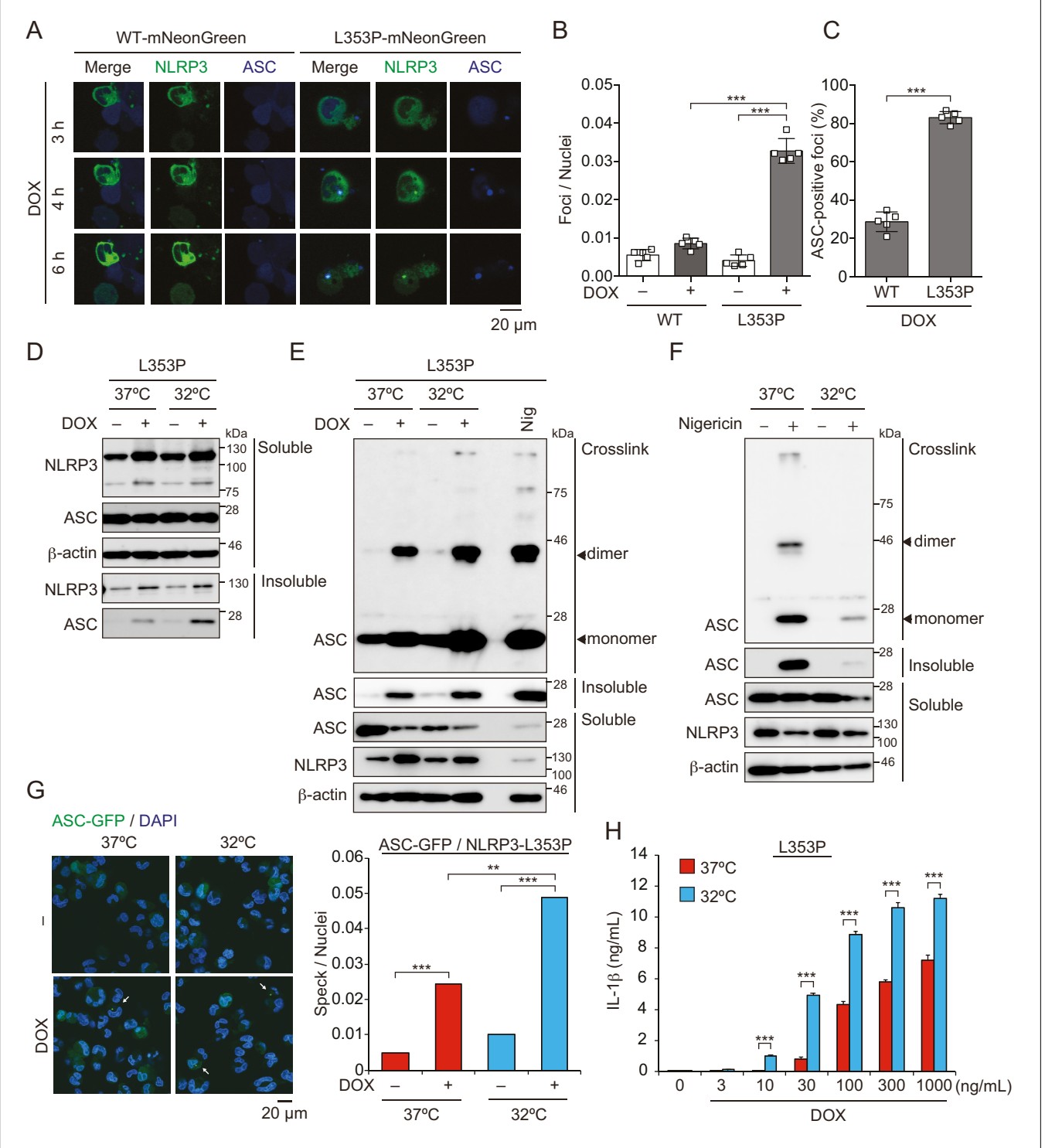

**Figure 3.** Aggregates formed by cryopyrin-associated periodic syndrome-associated NLRP3 mutant are the scaffold for inflammasome activation. (**A–C**) *EF1-ASC-BFP/TRE-NLRP3-WT* or *L353P-mNeonGreen*-THP-1 cells were treated with doxycycline (DOX). (**A**) Localization of apoptosis-associated speck-like protein containing a caspase recruitment domain (ASC) and NLRP3 was analyzed by confocal microscopy. (**B**) The number of foci was counted. (**C**) The ASC-speck number in NLRP3 foci was analyzed. (**D and E**) *TRE-NLRP3-L353P*-THP-1 cells were differentiated with phorbol-12-myristate-13-acetate (PMA) for 24 hr and then treated with DOX (30 ng/mL) at 37 or 32°C for 6 hr. (**D**) Triton X-soluble and triton X-insoluble fractions were analyzed by western blot. (**E**) Oligomerized ASC in Triton X-insoluble fractions was crosslinked with bis(sulfosuccinimidyl)suberate (BS₃) and analyzed by western blot. (**F**) Differentiated *TRE-NLRP3-L353P*-THP-1 cells were treated with nigericin at 37 or 32°C for 6 hr. Triton X-insoluble fractions were crosslinked with BS₃ and analyzed by western blot. (**G**) *EF-1-ASC-GFP/TRE-NLRP3-L353P*-THP-1 cells were differentiated with PMA for 24 hr and then treated with DOX

*Figure 3 continued on next page*

*Figure 3 continued*

(30 ng/mL) at 37 or 32°C for 6 hr. Representative images by confocal microscopy and the number of nuclei and specks were counted. (**H**) Differentiated *TRE-NLRP3-L353P*-THP-1 cells were treated with DOX at 37 or 32°C for 6 hr. The IL-1β levels in the supernatants were assessed by ELISA (n=3). (**B, C and H**) Data are expressed as the mean ± SD. **p<0.01 and ***p<0.005 as determined by (**B, C, and H**) two-way ANOVA with a post hoc test or (**G**) Fisher's exact test with the Holm correction. Data are representative of two or three independent experiments.

The online version of this article includes the following source data and figure supplement(s) for figure 3:

**Source data 1.** Source data for *Figure 3D*.

**Source data 2.** Source data for *Figure 3E*.

**Source data 3.** Source data for *Figure 3F*.

**Source data 4.** Source data for *Figure 3G*.

**Figure supplement 1.** Cold exposure enhances IL-1β release in NLRP3 familial cold autoinflammatory syndrome mutant-expressing cells.

**Figure supplement 1—source data 1.** Source data for *Figure 3—figure supplement 1A*.

**Figure supplement 1—source data 2.** Source data for *Figure 3—figure supplement 1B*.

**Figure supplement 1—source data 3.** Source data for *Figure 3—figure supplement 1C*.

**Figure supplement 1—source data 4.** Source data for *Figure 3—figure supplement 1F*.

$Ca^{2+}$ levels were not provoked by membrane rupture because $Ca^{2+}$ influx occurred prior to the release of cytosolic content as indicated by Kusabira orange or membrane permeabilization as indicated by SYTOX (*Figure 6—figure supplement 2A-D*). Notably, NLRP3-L353P-mediated inflammasome assembly monitored by ASC-BFP and $Ca^{2+}$ increase occurred coincidentally (*Figure 6G–I*). The size of ASC speck increased with the increase in $Ca^{2+}$. These results indicate that $Ca^{2+}$ influx occurs during inflammasome activation induced by NLRP3 mutants.

## Caspase-1 inhibition prevents FCAS-associated NLRP3 mutant-mediated inflammasome assembly

Recent studies have suggested that caspase activation causes the influx of $Ca^{2+}$ (*Rühl et al., 2018*). Therefore, we postulated that caspase-dependent $Ca^{2+}$ influx might enhance inflammasome assembly. To investigate the effect of caspase-1 inhibition on $Ca^{2+}$ influx, cells were treated with VX-765, a caspase-1 inhibitor, prior to DOX-mediated NLRP3-L353P induction. Indeed, VX-765 canceled $Ca^{2+}$ influx (*Figure 7A–C*, *Video 2*, *Figure 7—figure supplement 1A and B*) and inhibited ASC oligomerization and speck formation induced by NLRP3-L353P (*Figure 7D and E*). In accordance with reduced inflammasome assembly, VX-765 also prevented mutated NLRP3-induced IL-1β release (*Figure 7F* and *Figure 7—figure supplement 1* C). To further validate the role of caspase-1 in $Ca^{2+}$ influx, we developed *CASP1*-deficient cells (*Figure 7—figure supplement 1D*). $Ca^{2+}$ influx and ASC-speck formation induced by NLRP3-L353P were attenuated by *CASP1* deficiency (*Figure 7G–I*). These results indicate that caspase-1 activation induced by NLRP3 mutants promotes incremental inflammasome assembly by regulating $Ca^{2+}$ influx.

## Pannexin 1 inhibition attenuates FCAS-associated NLRP3 mutant-mediated inflammasome assembly

Pannexin 1, a large-pore channel, is activated by caspases and functions as a $Ca^{2+}$-permeable channel. Therefore, we investigated the contribution of pannexin 1 to mutated NLRP3-mediated inflammasome activation. A pannexin 1 inhibitor, trovafloxacin, attenuated the $Ca^{2+}$ influx and ASC-speck formation induced by NLRP3-L353P (*Figure 8A–F*). Moreover, probenecid, which is a clinically used drug for gout and inhibits pannexin 1, attenuated ASC-speck formation. Since probenecid was used in fluo-8 imaging to block fluo-8 leakage, we confirmed that caspase-1-dependent $Ca^{2+}$ influx was induced in the absence of probenecid (*Figure 8—figure supplement 1A-D*). Finally, the inhibition of pannexin 1 by trovafloxacin and probenecid-attenuated IL-1β release induced by NLRP3-L353P, whereas it was not effective in preventing nigericin-induced IL-1β release (*Figure 8G and H*, *Figure 8—figure supplement 1E and F*). However, the effect of pannexin 1 inhibition is limited to FCAS-associated mutants because it failed to inhibit IL-1β release induced by NLRP3-D303N (*Figure 8—figure supplement 1G*). On the other hand, MCC950 (*Coll et al., 2015*), a potent NLRP3 inhibitor, failed to prevent ASC-speck formation and IL-1β release induced by NLRP3-L353P, although MCC950 efficiently inhibited

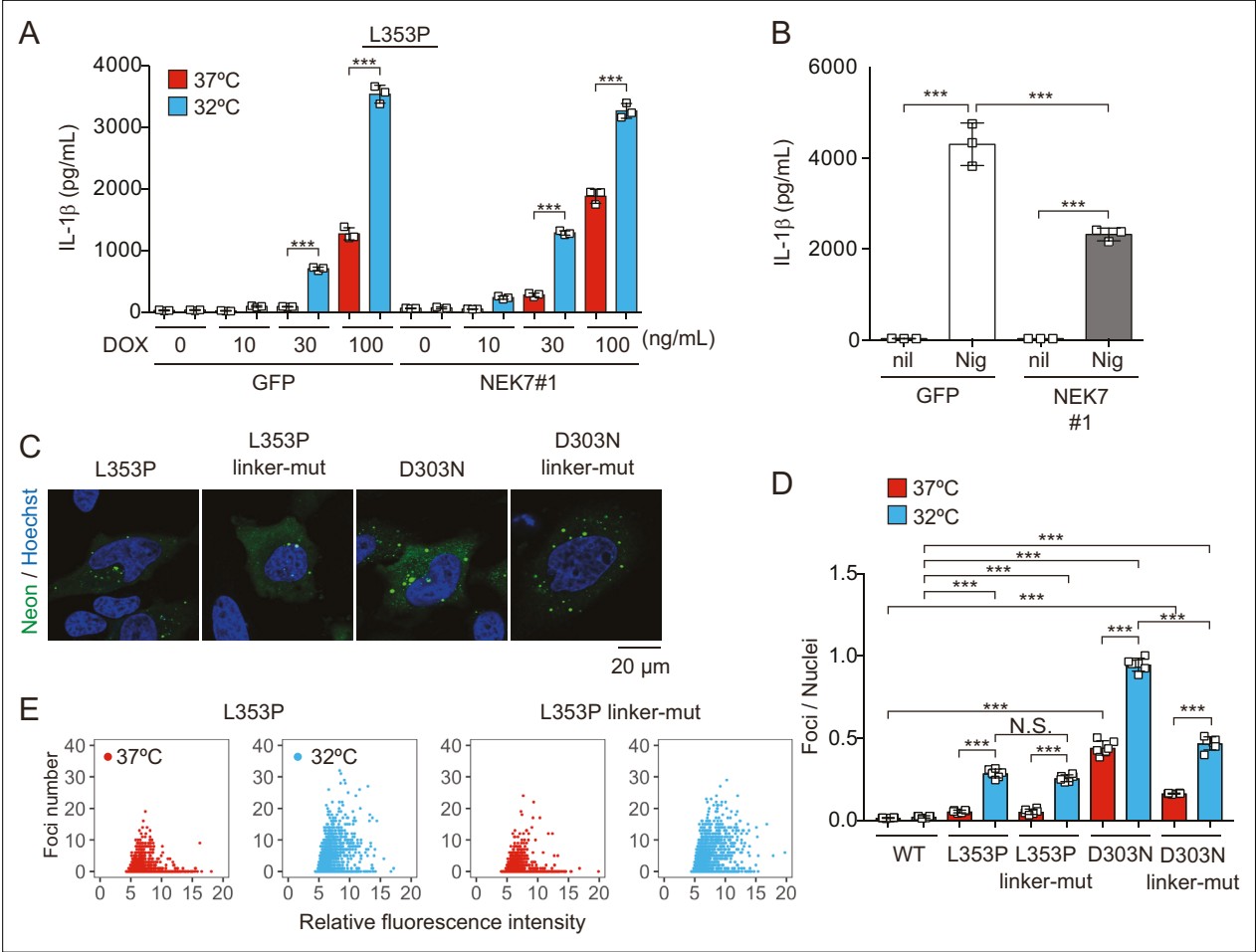

**Figure 4.** NEK7 and polybasic linker in NLRP3 are dispensable for cryopyrin-associated periodic syndrome-associated NLRP3 mutant-mediated inflammasome assembly. (**A and B**) Differentiated NEK7-mutated *TRE-NLRP3-L353P*-THP-1 cells were treated with (**A**) doxycycline (DOX) (30 ng/mL) or (**B**) nigericin at 37 or 32°C for 6 hr. The levels of IL-1β in the supernatants were assessed by ELISA (n=3). (**C**) *EF1-NLRP3-L353P-*, *NLRP3-L353P3KA-*, *EF1-NLRP3-D303N-*, or *NLRP3-D303N3KA-mNeonGreen*-HeLa cells were analyzed by confocal microscopy. (**D and E**) *EF1-NLRP3-L353P-*, *NLRP3-L353P-linker mutant-*, *EF1-NLRP3-D303N-*, or *NLRP3-D303N-linker mutant-mNeonGreen*-HeLa cells were cultured at 37 or 32°C for 24 hr. The number of foci and the fluorescence intensity of the cells were analyzed by high-content analysis. Data are expressed as the mean ± SD. ***p<0.005 as determined by two-way ANOVA with a post hoc test. Data are representative of three independent experiments.

The online version of this article includes the following source data and figure supplement(s) for figure 4:

**Source data 1.** Source data for *Figure 4E*.

**Figure supplement 1.** K⁺ efflux is dispensable for inflammasome activation induced by cryopyrin-associated periodic syndrome-associated NLRP3 mutant.

**Figure supplement 1—source data 1.** Source data for *Figure 4—figure supplement 1C*.

**Figure supplement 1—source data 2.** Source data for *Figure 4—figure supplement 1F*.

nigericin-induced IL-1β release (*Figure 8I and J*, *Figure 8—figure supplement 1H*). These results suggest that inhibition of pannexin 1-mediated $Ca^{2+}$ influx could be a potential therapeutic target of FCAS.

## Discussion

In the present study, we demonstrated that CAPS-associated NLRP3 mutants form cryo-sensitive foci intracellularly. These foci are aggregates that function as a scaffold for inflammasome activation. Consistent with this finding, inflammasome assembly and subsequent IL-1β release induced by CAPS-associated NLRP3 mutants are cryo-sensitive. The aggregation of CAPS-associated NLRP3 mutants

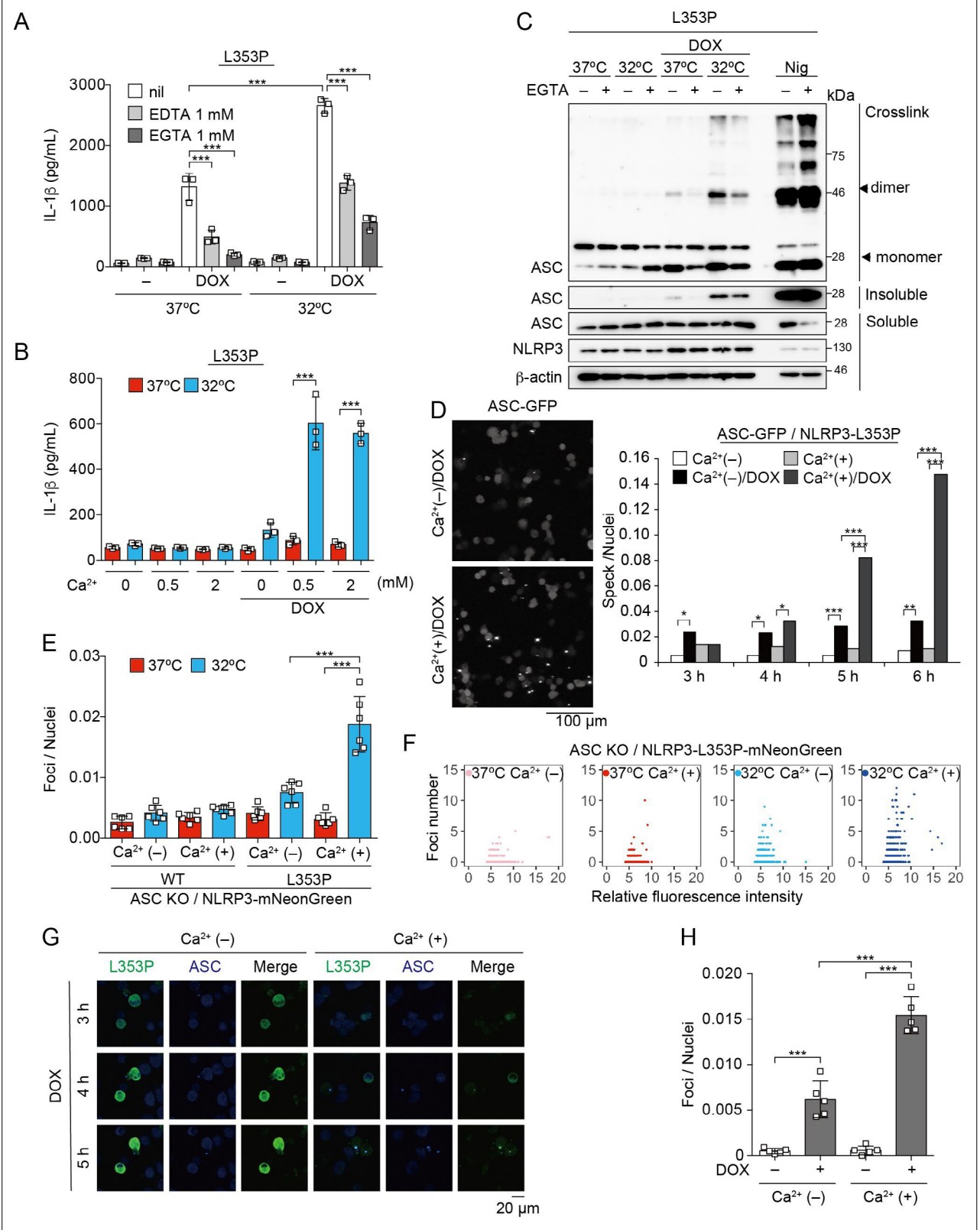

**Figure 5.** Ca²⁺ is required for cryopyrin-associated periodic syndrome-associated NLRP3 mutant-mediated inflammasome assembly. (**A–C**) Differentiated *TRE-NLRP3-L353P*-THP-1 cells were pretreated with indicated conditions, and then treated with doxycycline (DOX) (30 ng/mL) at 37 or 32°C for 6 hr. (**A and B**) The IL-1β levels in the supernatants were assessed by ELISA (n=3). (**C**) Oligomerized apoptosis-associated speck-like protein containing a caspase recruitment domain (ASC) in Triton X-insoluble fractions was crosslinked with bis(sulfosuccinimidyl)suberate and analyzed by

*Figure 5 continued on next page*

*Figure 5 continued*

western blot. (**D**) *EF1-ASC-GFP/TRE-NLRP3-L353P*-THP-1 cells were pretreated with $Ca^{2+}$-depleted or -supplemented media and then treated with DOX (30 ng/mL) at 37°C. ASC-speck formation was analyzed by confocal microscopy. (**E and F**) *ASC KO / EF1-NLRP3-WT-* or *L353P- mNeonGreen*-THP-1 cells were cultured at 37 or 32°C for 24 hr in $Ca^{2+}$-depleted or -supplemented media. The number of foci and fluorescent intensity was analyzed by high-content analysis. (**G and H**) Differentiated *EF1-ASC-BFP/TRE-NLRP3-L353P-mNeonGreen* THP-1 cells were pretreated with $Ca^{2+}$-depleted or -supplemented media and then treated with DOX (30 ng/mL) at 37°C. (**G**) Representative images by confocal microscopy. (**H**) The number of foci was analyzed by high-content analysis. (**A, B, F, and G**) Data are expressed as the mean ± SD. *p<0.05, **p<0.01, and ***p<0.005 as determined by (**A, B, E, and H**) two-way ANOVA with a post hoc test or (**D**) Fisher's exact test with the Holm correction. Data are representative of two or three independent experiments.

The online version of this article includes the following source data and figure supplement(s) for figure 5:

**Source data 1.** Source data for *Figure 4C*.

**Source data 2.** Source data for *Figure 4D*.

**Source data 3.** Source data for *Figure 5F* and *Figure 5—figure supplement 2*.

**Figure supplement 1.** $Ca^{2+}$ is necessary for inflammasome activation induced by cryopyrin-associated periodic syndrome-associated NLRP3 mutant.

**Figure supplement 1—source data 1.** Source data for *Figure 5—figure supplement 1C*.

**Figure supplement 1—source data 2.** Source data for *Figure 5—figure supplement 1D*.

**Figure supplement 2.** $Ca^{2+}$ is necessary for aggregation of cryopyrin-associated periodic syndrome-associated NLRP3 mutant.

is regulated by intracellular $Ca^{2+}$ levels. Furthermore, caspase-1 inhibition prevents $Ca^{2+}$ influx and inflammasome assembly induced by CAPS-associated NLRP3 mutants. These findings provide new insights into the molecular mechanisms of inflammasome activation in CAPS.

The formation of cryo-sensitive aggregates by CAPS-associated NLRP3 mutants is a significant finding of this study. Oligomerization or polymerization is a common feature of domains contained in inflammasome components. Both caspase recruitment domain (CARD) in ASC and caspase-1 and PYD in ASC and NLRP3 form filamentous assemblies (*Masumoto et al., 1999*; *Lu et al., 2014*; *Karasawa et al., 2015*; *Stutz et al., 2017*). In addition, full-length ASC, which has both PYD and CARD, forms a large aggregate called speck. Therefore, aggregation of CAPS-associated NLRP3 mutants seems to be mediated by their PYD-PYD interaction. However, the mechanisms by which CAPS-associated NLRP3 mutants, which typically occur in other domains including NACHT domain and LRR, promote aggregation and affect their cryo-sensitivity have remained unclear. During the preparation of this manuscript, the structure of full-length NLRP3 and complex of NLRP3 oligomer have been determined. *Andreeva et al., 2021* have suggested that full-length murine NLRP3 forms oligomer called double-ring cage via interaction with LRR. Further analyses are required to clarify the role of double-ring cage formation in aggregation of CAPS-associated NLRP3 mutants.

With regard to cryo-sensitivity, recent studies have suggested that the formation of aggregates and LLPS is modulated by conditions including temperature and pH (*Alberti et al., 2019*; *Riback et al., 2020*). These factors shift the threshold of aggregation and LLPS under a constant protein concentration. In the present study, the frequency of foci formation by mutated NLRP3 was weakly associated with its expression levels. Furthermore, increased expression of mutated NLRP3 dose dependently promoted inflammasome assembly. The temperature and its expression levels seems to be an essential determinant of the aggregation of mutated NLRP3. Although all of the analyzed NLRP3 mutants exhibited cryo-sensitivity, the sensitivity of FCAS-associated L353P and Y563N mutants was more prominent than that of CINCA-associated D303N and Y570C mutants. Meanwhile, CINCA-associated mutants formed a large number of aggregates compared to FCAS-associated mutants at 37°C. Therefore, we assume that different cryo-sensitivity plays a causative role in the disease severity and sensitivity to cold exposure of CAPS. Further analyses are required to clarify the association between disease severity and capacity to form aggregates. Cold exposure also affected inflammasome activation induced by extrinsic stimulation. Unlike NLRP3 mutant-mediated inflammasome activation, nigericin- or nanosilica-induced NLRP3 inflammasome activation was blunted by cold exposure. Since NLRP3 inflammasome assembly requires ATPase activity, the temperature may influence extrinsic stimulation-induced inflammasome assembly via its enzymatic activity (*Duncan et al., 2007*).

Increasing evidence suggests that $K^+$ efflux is a main upstream event of NLRP3 inflammasome activation (*Munoz-Planillo et al., 2013*; *He et al., 2016a*). Although the precise mechanism underlying $K^+$ efflux-mediated NLRP3 inflammasome assembly is still unclear, NEK7 has been shown to

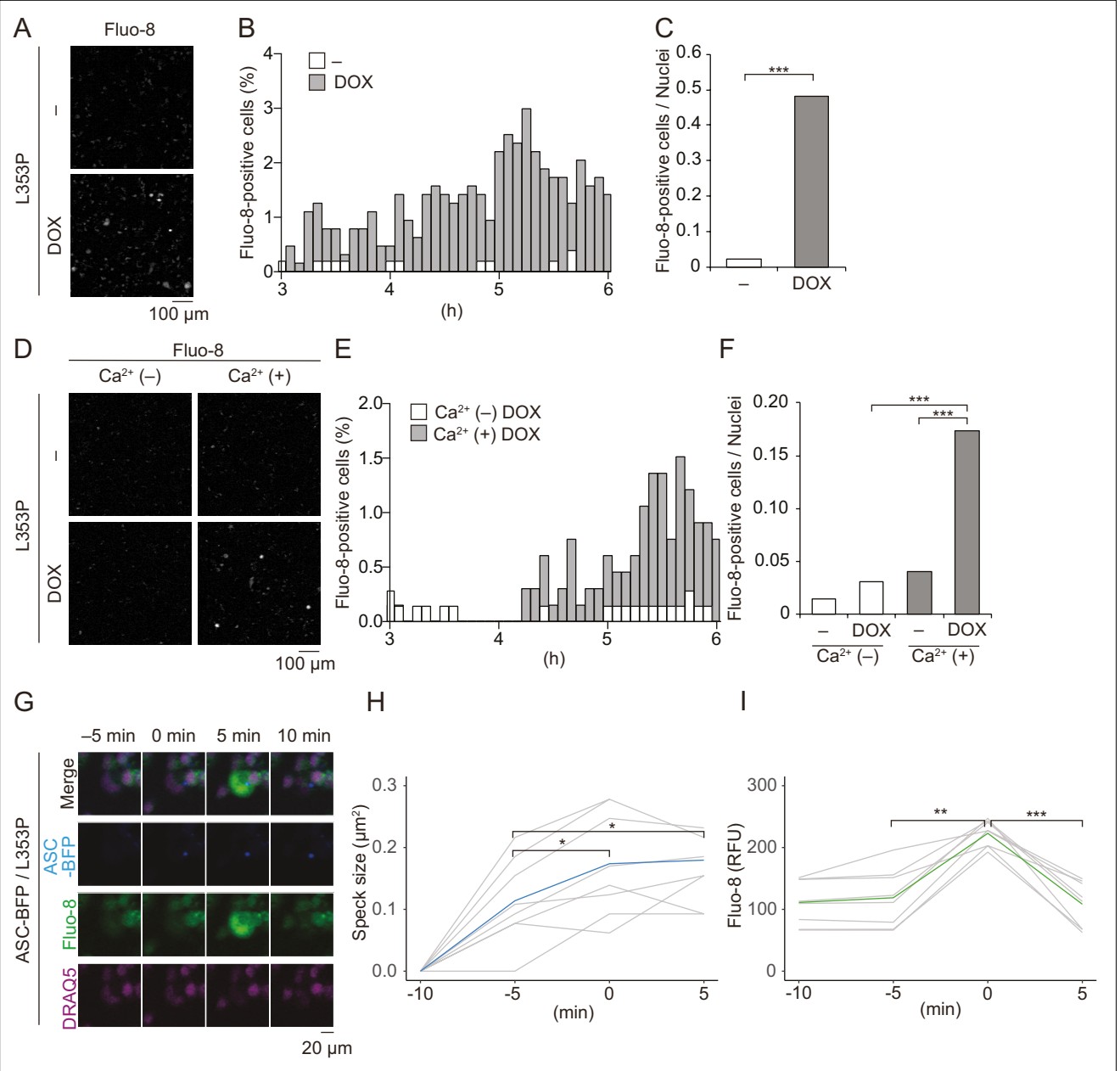

**Figure 6.** Ca²⁺ influx is provoked during mutated NLRP3-mediated inflammasome assembly. (**A–C**) Differentiated *TRE-NLRP3-L353P*-THP-1 cells were loaded with 4 µM Fluo-8 for 1 hr and treated with doxycycline (DOX) (30 ng/mL) at 37°C for 6 hr. The images were recorded by confocal microscopy at 5 min intervals from 3 hr to 6 hr. (**A**) Representative temporal subtraction images. (**B**) The frequency of intracellular Ca²⁺ increase at each time point. (**C**) The cumulative number of Fluo-8-positive cells. (**D–F**) Differentiated *TRE-NLRP3-L353P*-THP-1 cells were loaded with 4 µM Fluo-8 for 1 hr and treated with DOX (30 ng/mL) at 37°C for 6 hr in Ca²⁺-depleted or -supplemented media. The images were recorded by confocal microscopy at 5 min intervals from 3 hr to 6 hr. (**D**) Representative temporal subtraction images. (**E**) The frequency of intracellular Ca²⁺ increase at each time point. (**F**) The cumulative number of Fluo-8-positive cells. (**G–I**) Differentiated *EF1-ASC-BFP/TRE-NLRP3-L353P*-THP-1 cells were loaded with 4 µM Fluo-8 for 1 hr and treated with DOX (30 ng/mL) at 37°C. The images were recorded at 5 min intervals. (**G**) Representative images of the cells with increased Fluo-8 signals. (**H**) The apoptosis-associated speck-like protein containing a caspase recruitment domain (ASC)-BFP speck size (**I**) and fluorescent intensity of Fluo-8 were analyzed. The peak time point of Fluo-8 signals was defined as 0 min. (**H**) The blue line and the (**I**) green line represent mean values, and the gray line represents each measurement. *p<0.05, **p<0.01, and ***p<0.005 as determined by (**C and F**) Fisher's exact test with the Holm correction or (**H and I**) repeated one-way ANOVA with a post hoc test. (**A–G**) Data are representative of three independent experiments. (**H and I**) Data are from three independent live-cell imaging.

The online version of this article includes the following source data and figure supplement(s) for figure 6:

**Source data 1.** Source data for *Figure 6B and C*.

*Figure 6 continued on next page*

*Figure 6 continued*

**Source data 2.** Source data for *Figure 6E and F*.

**Figure supplement 1.** Ca²⁺ influx is induced by chronic infantile neurological, cutaneous, and articular syndrome-associated NLRP3 mutant.

**Figure supplement 1—source data 1.** Source data for *Figure 6—figure supplement 1A*.

**Figure supplement 2.** Pyroptosis is induced by mutated NLRP3 after Ca²⁺ influx.

interact with NLRP3 directly and promote inflammasome assembly as a downstream of $K^+$ efflux (*He et al., 2016b*). Unexpectedly, we demonstrated that inflammasome activation induced by NLRP3-L353P is not prevented by supplementation with extracellular $K^+$ or a deficiency of NEK7, indicating that $K^+$ efflux is dispensable for inflammasome activation induced by FCAS-associated NLRP3 mutant. Instead, our data clearly showed that both NLRP3 aggregation and inflammasome assembly induced by NLRP3-L353P are dependent on the presence of $Ca^{2+}$. Several studies indicate that $Ca^{2+}$ is another upstream signal of NLRP3 inflammasome activation (*Lee et al., 2012*; *Murakami et al., 2012*; *Rossol et al., 2012*; *Horng, 2014*). Further studies are necessary to elucidate whether the previously identified mechanism of $Ca^{2+}$-mediated WT-NLRP3 inflammasome activation shares the mechanism with inflammasome activation induced by CAPS-associated NLRP3 mutants. However, the $Ca^{2+}$-sensitive NLRP3 aggregation appears to be distinct from canonical inflammasome assembly because the stimulation with nigericin failed to form NLRP3 aggregates.

Of note, we showed that the elevation of intracellular $Ca^{2+}$ induced by NLRP3-L353P was dependent on caspase-1 activity. Recent studies have suggested that gasdermin pore formation induced by caspase activation regulates the influx of $Ca^{2+}$ (*de Vasconcelos et al., 2019*). Broz and his colleagues have suggested that the $Ca^{2+}$ influx induced by GSDMD pore formation initiates membrane repair by ESCRT complex, which negatively regulates pyroptosis (*Rühl et al., 2018*). Another possible pore channel downstream of caspases is pannexin 1. Previous studies have suggested that caspase-11 and caspase-7 cleave and activate pannexin 1 during pyroptosis and apoptosis, respectively (*Chekeni et al., 2010*; *Yang et al., 2015*). The $Ca^{2+}$ influx induced by NLRP3-L353P was probably mediated by pannexin 1 because two pannexin 1 inhibitors, probenecid and trovafloxacin, attenuated $Ca^{2+}$ influx and inflammasome assembly induced by NLRP3-L353P. In the present study, pharmacological inhibition of caspase-1 and pannexin 1 prevents the elevation of intracellular $Ca^{2+}$ and ASC oligomerization and speck formation. Therefore, we consider that CAPS-associated NLRP3 mutant forms cryo-sensitive inflammasome assemblies, which trigger caspase-1-mediated feed-forward loop of $Ca^{2+}$ influx, leading to incremental inflammasome assembly. $Ca^{2+}$ seems to promote this incremental inflammasome assembly rather than the initial assembly because the aggregation of NLRP3 mutants was blunted but still induced in the absence of $Ca^{2+}$. In comparison with NLRP3-L353P, NLRP3-D303N was less sensitive to $Ca^{2+}$ because $Ca^{2+}$ depletion partially inhibited IL-1β release induced by NLRP3-D303N. Furthermore, the pannexin 1 inhibition failed to prevent IL-1β release induced by NLRP3-D303N. It is likely that CINCA-associated mutants exhibit constitutive active properties, and feed-forward regulation mediated by $Ca^{2+}$ influx promotes inflammasome assembly only in FCAS-associated NLRP3 mutants. Taken together, the inhibition of caspase-1-pannexin 1 axis could be a potential target to inhibit inflammasome assembly induced by FCAS-associated NLRP3 mutants.

In the present study, we investigated the mechanisms of cryo-sensitive inflammasome assembly

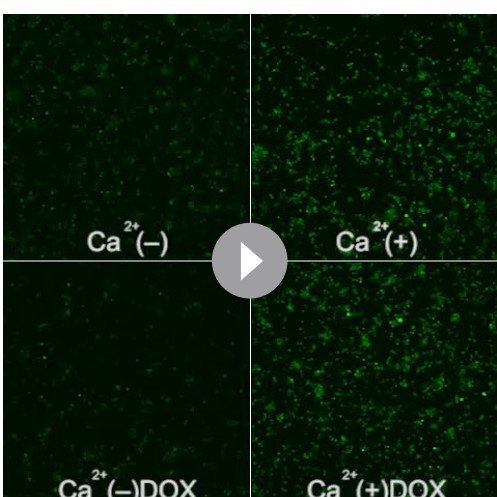

**Video 1.** Ca²⁺ influx is induced by familial cold autoinflammatory syndrome-associated NLRP3 mutant. Differentiated *TRE-NLRP3-L353P-THP-1* cells were loaded with 4 µM Fluo-8 for 1 hr and treated with doxycycline (30 ng/mL) at 37°C for 6 hr in Ca²⁺-depleted or -supplemented media. The images were recorded by confocal microscopy at 5 min intervals from 3 hr to 6 hr.

https://elifesciences.org/articles/75166/figures#video1

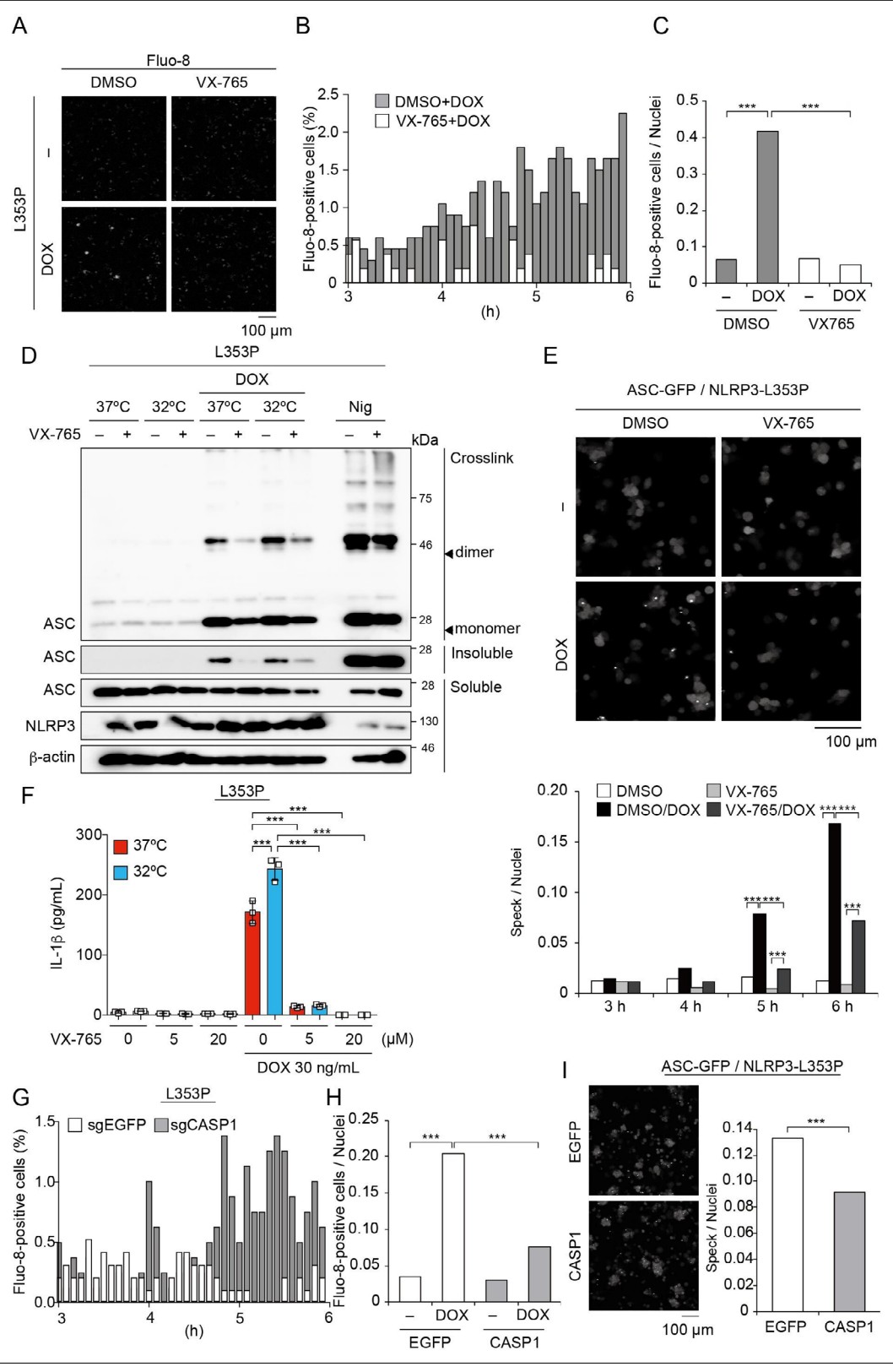

**Figure 7.** Caspase-1 inhibition prevents familial cold autoinflammatory syndrome-associated NLRP3 mutant-mediated inflammasome assembly. (**A –C**) Differentiated *TRE-NLRP3-L353P*-THP-1 cells were loaded with 4 µM Fluo-8 for 1 hr and were pretreated with VX-765 (20 µM) for 30 min. After doxycycline (DOX) (30 ng/mL) treatment, the images were recorded at 5 min intervals from 3 hr to 6 hr. (**A**) Representative temporal subtraction images.

*Figure 7 continued on next page*

*Figure 7 continued*

(**B**) The frequency of intracellular $Ca^{2+}$ increase at each time point. (**C**) The cumulative number of Fluo-8-positive cells. (**D and F**) Differentiated *TRE-NLRP3-L353P*-THP-1 cells were pretreated with VX-765 (20 μM) for 30 min and then treated with DOX (30 ng/mL) or nigericin (5 μM) at 37 or 32°C. (**D**) Triton X-insoluble fractions were crosslinked with bis(sulfosuccinimidyl)suberate and analyzed by western blot. (**F**) The IL-1β levels in the supernatants were assessed by ELISA (n=3). (**E**) *EF1-ASC-GFP/TRE-NLRP3-L353P*-THP-1 cells were pretreated with 20 μM VX-765 for 30 min and then treated with DOX (30 ng/mL) at 37°C. ASC-speck formation was analyzed by confocal microscopy. (**G and H**) The differentiated CASP1-mutated *TRE-NLRP3-L353P*-THP-1 cells were treated with 4 μM Fluo-8 for 1 hr and treated with DOX (30 ng/mL) at 37°C for 6 hr. (**G**) The frequency of intracellular $Ca^{2+}$ increase at each time point. (**H**) The cumulative number of Fluo-8-positive cells. (**I**) The differentiated CASP1-mutated *EF1-ASC-GFP/TRE-NLRP3-L353P*-THP-1 cells were treated with DOX (30 ng/mL) at 37°C for 6 hr. ASC-speck formation was analyzed by confocal microscopy. (**F**) Data are expressed as the mean ± SD. *$p<0.05$, **$p<0.01$, and ***$p<0.005$ as determined by (**C, E, G, and H**) Fisher's exact test with the Holm correction or (**F**) two-way ANOVA with a post hoc test. Data are representative of two or three independent experiments.

The online version of this article includes the following source data and figure supplement(s) for figure 7:

**Source data 1.** Source data for *Figure 7B and C*.

**Source data 2.** Source data for *Figure 7D*.

**Source data 3.** Source data for *Figure 7E*.

**Source data 4.** Source data for *Figure 7G*.

**Source data 5.** Source data for *Figure 7H*.

**Figure supplement 1.** Caspase activity is required for inflammasome assembly induced by familial cold autoinflammatory syndrome-associated NLRP3 mutant.

**Figure supplement 1—source data 1.** Source data for *Figure 7—figure supplement 1C*.

in CAPS-associated NLRP3 mutants. However, this study has a few limitations. First, to analyze the function of mutated NLRP3, we used mutated NLRP3 expressing under an inducible promoter or in *ASC KO* cells because the expression of mutated NLRP3 induces pyroptotic cell death. To clarify the dynamics of endogenously expressed NLRP3, a study using a knock-in model would be required. Second, the aggregation of NLRP3 was analyzed by fusion protein with mNeonGreen. A biochemical analysis of aggregation using mutated NLRP3 without any tags or fluorescent proteins is also required in the future.

In conclusion, we found that CAPS-associated NLRP3 mutants form cryo-sensitive aggregates, which can function as the scaffold for NLRP3 inflammasome activation. The aggregation of mutated NLRP3 is not dependent on $K^+$ efflux but rather is regulated by intracellular $Ca^{2+}$ levels. We expect that our findings would be valuable for the development of novel therapies for CAPS.

**Video 2.** Caspase activity is required for $Ca^{2+}$ influx induced by familial cold autoinflammatory syndrome-associated NLRP3 mutant. Differentiated *TRE-NLRP3-L353P-THP-1* cells were loaded with 4 μM Fluo-8 for 1 hr and were pretreated with VX-765 (20 μM) for 30 min. After doxycycline (30 ng/mL) treatment, the images were recorded at 5 min intervals from 3 hr to 6 hr. https://elifesciences.org/articles/75166/figures#video2

# Materials and methods
## Plasmids

PCR-generated cDNAs encoding human NLRP3 were subcloned into pENTR4 vector. The mutated NLRP3-D303N and NLRP3-L353P were generated using the PrimeSTAR Mutagenesis Basal kit (Takara Bio, Shiga, Japan) (*Aizawa et al., 2020*). The primers for introducing mutations were as follows: D303N (forward, 5'-GGCT TCAATGAGCTGCAAGGTGCCTTTGACGAG-3'; reverse, 5'-CAGCTCATTGAAGCCGTCCATGAG GAAGAGGAT-3') and L353P (forward, GTGG CCCCGGAGAAACTGCAGCACTTGCTGGAC-3';

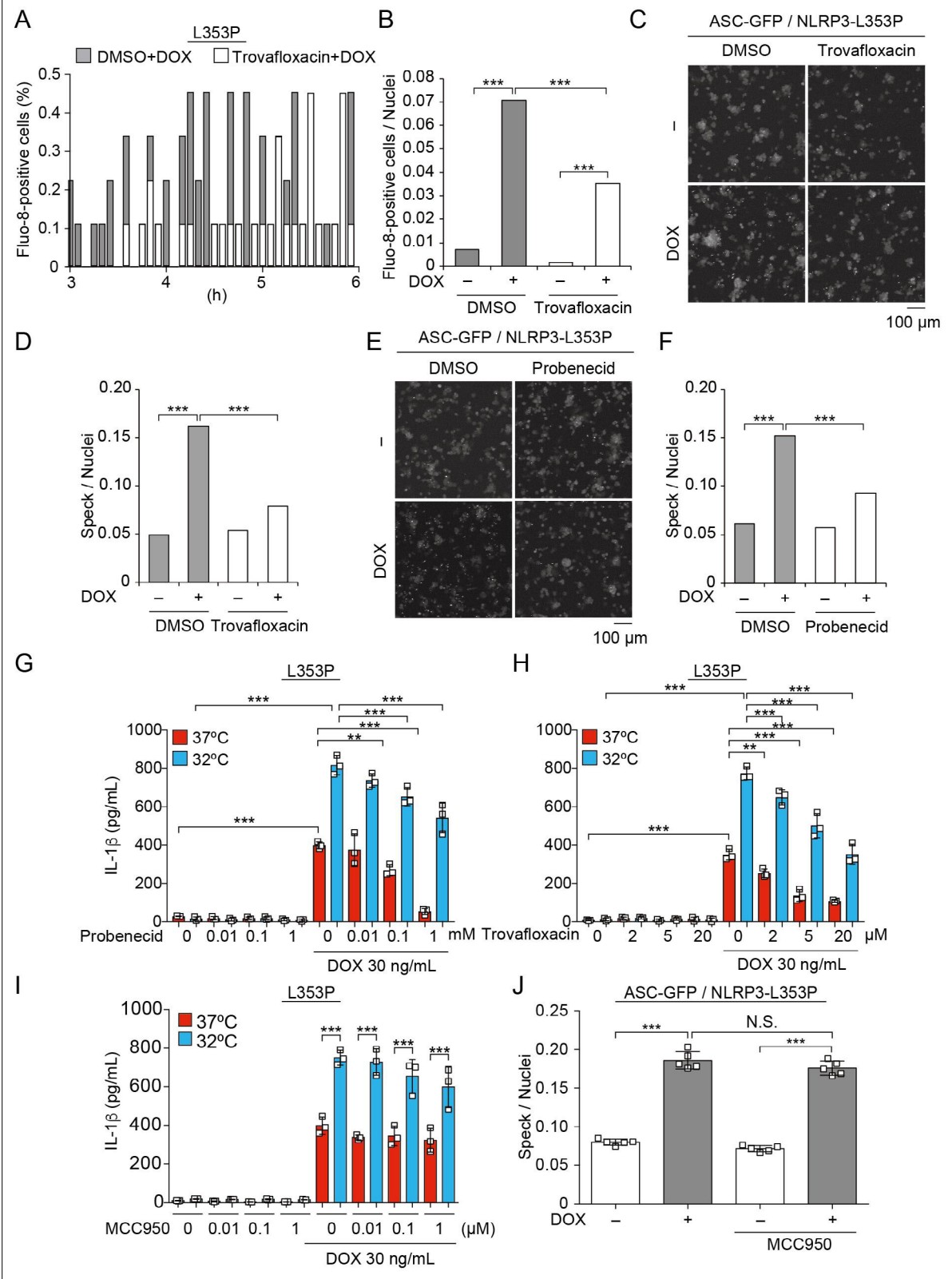

**Figure 8.** Pannexin 1 inhibition prevents familial cold autoinflammatory syndrome-associated NLRP3 mutant-mediated inflammasome assembly. (**A and B**) Differentiated *TRE-NLRP3-L353P*-THP-1 cells were pretreated with 4 μM Fluo-8 for 1 hr and trovafloxacin (20 μM) for 30 min. After doxycycline (DOX) (30 ng/mL) treatment, the images were recorded at 5 min intervals from 3 hr to 6 hr. (**A**) The frequency of intracellular Ca²⁺ increase at each time point. (**B**) The cumulative number of Fluo-8-positive cells. (**C–F**) *EF1-ASC-GFP/TRE-NLRP3-L353P*-THP-1 cells were pretreated with (**C and**

*Figure 8 continued on next page*

*Figure 8 continued*

D) trovafloxacin (20 µM) or (**E and F**) probenecid (1 mM) for 30 min and then treated with DOX (30 ng/mL) at 37°C for 6 hr. Apoptosis-associated speck-like protein containing a caspase recruitment domain (ASC)-speck formation was analyzed by confocal microscopy. (**C and E**) Representative images by confocal microscopy. (**D and F**) The number of nuclei and specks was counted. (**G – I**) Differentiated *TRE-NLRP3-L353P*-THP-1 cells were pretreated with (**G**) trovafloxacin, (**H**) probenecid, or (**I**) MCC950 for 30 min and then treated with DOX (30 ng/mL) at 37 or 32°C for 6 hr. The IL-1β levels in the supernatants were assessed by ELISA (n=3). (**J**) *EF1-ASC-GFP/TRE-NLRP3-L353P*-THP-1 cells were pretreated with MCC950 and then treated with DOX (30 ng/mL) at 37°C. The formation of ASC speck was analyzed by high-content analysis. (**G–J**) Data are expressed as the mean ± SD. \*\*$p<0.01$ and \*\*\*$p<0.005$ as determined by (**G–J**) two-way ANOVA with a post hoc test or (**B, D, and F**) Fisher's exact test with the Holm correction. Data are representative of three independent experiments.

The online version of this article includes the following source data and figure supplement(s) for figure 8:

**Source data 1.** Source data for *Figure 8A and B*.

**Source data 2.** Source data for *Figure 8D*.

**Source data 3.** Source data for *Figure 8F*.

**Figure supplement 1.** Pannexin 1 inhibition prevents familial cold autoinflammatory syndrome-associated NLRP3 mutant-mediated inflammasome assembly.

**Figure supplement 1—source data 1.** Source data for *Figure 8—figure supplement 1A*.

**Figure supplement 1—source data 2.** Source data for *Figure 8—figure supplement 1C*.

reverse, 5′-TTTCTCCGGGGCCACAGGTCTCGTGGTGATGAG-3′). To produce DOX-inducible expression vector, the mutated NLRP3 and WT-NLRP3 were transferred into CS-IV TRE CMV KT (kindly provided by Dr. H. Miyoshi, RDB12876, RIKEN BRC, Tsukuba, Japan) with LR clonase (Thermo Fisher Scientific). To develop NLRP3-mNeonGreen or ASC-BFP-expressing lentiviral vector, PCR-generated NLRP3-WT, NLRP3-L353P, NLRP3-D303N, NLRP3-Y563N, NLRP3-Y570C, NLRP3-L353P3KA, NLRP3-D303N3KA, ASC, mNeonGreen, and moxBFP (a gift from Erik Snapp; Addgene plasmid #68064) were Gibson subcloned into CS-EF-1 (derived from CS-CA-MCS; RIKEN BRC). The sgRNA targeting NEK7 was designed with CRISPR direct (http://crispr.dbcls.jp) and subcloned into LentiCRISPRv2, which was a gift from Feng Zhang (Addgene plasmid #52961; http://n2t.net/addgene: 52961; RRID: Addgene_52961). The sgRNA targeting ASC and CASP1 was developed previously (*Aizawa et al., 2020*).

## Cell lines

HeLa cells (gifted from Dr. Kenji Tago) were cultured in Dulbecco's modified Eagle's medium (DMEM, Wako, Osaka, Japan) supplemented with 10% fetal calf serum (FCS) and antibiotics. THP-1 cells (ATCC) were cultured in RPMI1640 (Sigma, St Louis, MO, USA) supplemented with 10% FCS and antibiotics. THP-1 macrophages were differentiated with 200 nM phorbol-12-myristate-13-acetate (PMA) for 24 hr. LentiX293T cells were obtained from TAKARA (Takara Bio, Shiga, Japan) and cultured in DMEM supplemented with 10% FCS, 1 mM sodium pyruvate, and antibiotics. Unless otherwise indicated, cells were cultured at 37°C in 5% $CO_2$. Cell lines were authenticated by analysis of short tandem repeat profiling (BEX, Tokyo, Japan) and confirmed as negative for mycoplasma contamination using TaKaRa PCR Mycoplasma Detection Set (Takara Bio) and Hoechst staining.

## Lentiviral preparation

LentiX293T cells were co-transfected with self-inactivating vectors, pLP1, pLP2, and pVSVG using PEI MAX (Polysciences, Warrington, PA, USA) to prepare the lentiviral vectors. Culture media containing the lentiviral vectors were collected 3 days after transfection. The collected media were filtered with a 0.45 µm filter and ultracentrifuged at 21,000 rpm using a Type 45 Ti rotor (Beckman Coulter, Brea, CA, USA), and the pellets were resuspended in PBS containing 5% FCS. The lentivirus titer was measured using a Lentivirus quantitative PCR Titer kit (Applied Biological Materials, Richmond, BC, Canada). For lentiviral transduction, the cells were incubated with purified lentiviral vectors in the presence of 8 µg/mL polybrene (Sigma). The details of the developed cells are described in the key resource table.

## Treatment of reagents and cold exposure

The transduced THP-1 cells were differentiated with 200 nM PMA for 24 hr and treated with DOX (Wako) or nigericin (InvivoGen) at the indicated concentrations. Next, cells were cultured at 37 or

32°C. Cells were then pretreated with inhibitors including CA-074 (Wako), MCC950 (Adipogen, San Diego, CA), probenecid (Cayman, Ann arbor, MI), trovafloxacin mesylate (Cayman), VX-765 (Selleck), and Z-VAD-FMK (MBL) for 30 min prior to cold exposure.

## Confocal microscopy

For imaging of fixed cells, the transduced cells were seeded on an eight-well chamber slide (Matsunami Glass Ind., Ltd., Osaka, Japan) and then fixed with neutral buffered formalin or 1% paraformaldehyde and stained with 1 μg/mL 4',6-diamidino-2-phenylindole, dihydrochloride (DAPI; Dojindo). For live-cell imaging, cells were seeded at $1\times10^5$ cells/well on eight-well cover glass chambers (IWAKI, Shizuoka, Japan) and labeled with Hoechst33342 for 20 min before treatment. The images were captured using confocal microscopy (FLUOVIEW FV10i; Olympus, Tokyo, Japan).

## High-content analysis

The transduced HeLa cells or THP-1 cells were seeded on 96-well plates and treated with the indicated stimuli. For analysis of fixed cells, nuclei were stained with DAPI after fixation by 1% paraformaldehyde. For live-cell imaging, cells were stained with DRAQ5 (Biolegend, San Diego, CA) and analyzed by an Operetta CLS high-content analysis system (PerkinElmer, Waltham, MA).

## FRAP analysis

The transduced HeLa cells were seeded on eight-well cover glass chambers (IWAKI), cultured at 32°C for 24 hr, and analyzed by FV1000 confocal microscopy (Olympus) at 32°C. Images were captured using an UPLA SAPO 60XO objective. The GFP and mNeonGreen signals were captured using a line sequential scan setting with excitation laser lines at 488 nm. For FRAP analysis, a 1 s pulse of the 488 nm laser line at 5% power was used to bleach the NLRP3-mNeonGreen foci. A 5 s pulse of 488 nm laser line at 30% power was used to bleach the ASC-GFP specks, and a 1 s pulse of 405 nm laser line at 50% power was used to bleach unfused mNeonGreen. The changing of fluorescence was monitored by imaging every 0.2 ms.

## IL-1β secretion assay

Cells were seeded into 96-well plates at $5\times10^4$ cells/well. After the indicated treatments, culture supernatants were collected and the IL-1β levels were measured by ELISA using a commercial kit (R&D Systems, Minneapolis, MN, USA). The supernatants were precipitated with ice-cold acetone and resolved in 1×Laemmli buffer for western-blot analysis.

## Western-blot analysis

Samples were separated by sodium dodecyl sulfate-polyacrylamide electrophoresis and transferred to polyvinylidene difluoride (PVDF) membranes. After blocking with Blocking One (Nacalai Tesque, Kyoto, Japan) for 30 min, the membranes were incubated overnight at 4°C with the following primary antibodies (Abs): anti-ASC (AL-177; Adipogen), anti-β actin (clone AC-15; Sigma), anti-caspase-1 (3866; Cell Signaling Technology), anti-IL-1β (H153; Santa Cruz Biotechnology), anti-NEK7 (EPR4900; abcam, Cambridge, UK), and anti-NLRP3 (clone Cryo-2; Adipogen). As secondary Abs, HRP-goat anti-mouse Superclonal IgG (Thermo Fisher Scientific) or HRP-goat anti-rabbit IgG (Cell Signaling Technology) was incubated with membrane for 1 hr. After being washed with TBS-Tween, immunoreactive bands were visualized using Western BLoT Quant HRP Substrate (Takara Bio) or Western BLoT Ultra Sensitive HRP Substrate (Takara Bio).

## ASC-oligomerization assay

Cells were lysed in 0.5% Triton X-100 buffer (20 mM Tris HCl, 10 mM KCl, 1.5 mM MgCl$_2$, 1 mM EDTA, 1 mM EGTA, 320 mM sucrose, and 0.5% Triton X-100) for 20 min. Lysates were then centrifuged at 5000× g for 10 min. The insoluble pellets were reacted with 2 mM bis(sulfosuccinimidyl)suberate (Thermo Fisher Scientific) for 30 min and the reactions were terminated by an excess amount of glycine.

## Reverse transcription and real-time PCR

Total RNA was prepared using ISOGEN (Nippon Gene Co., Tokyo, Japan) according to the manufacturer's instructions. Total RNA was reverse transcribed using a SuperScript VILO cDNA Synthesis

kit (Life Technologies). Real-time PCR was performed using SYBR Premix Ex Taq II (Takara Bio). The primers used in the assay were as follows: NEK7 (forward, 5′-GCCTTACGACCGGATATGGG-3′; reverse, 5′-CACTAAATTGTCCGCGACCAA–3′) and ACTB (forward, 5′- GGCACTCTTCCAGCCTTCCT TC-3′; reverse, 5′-GCGGATGTCCACGTCACACTTCA-3′).

## Ca²⁺ imaging using Fluo-8

Cells were seeded at $1 \times 10^5$ cells/well on an eight-well cover glass chamber (IWAKI). Fluo-8 (Santa Cruz Biotechnology) was loaded at 4 µM for 60 min in the presence of 0.04% Pluronic F127 (Sigma) and 1.25 mM probenecid (Cayman). After removal of the loading medium, cells were treated with DOX in the presence of 1.25 mM probenecid. In experiments not using probenecid, cells were pretreated with Fluo-8 at 4 µM for 60 min in the presence of 0.04% Pluronic F127 and then treated with DOX. Z-stack time-lapse images at 37°C were captured using confocal microscopy (FLUOVIEW FV10). To normalize the cell number, nuclei were labeled with Hoechst 33,342 (1 µg/mL) or DRAQ5. For detection of dying cells, images were captured in the presence of 100 nM SYTOX Deep Red.

## Fura2 assay

Cells were seeded at $5 \times 10^4$ cells/well into 96-well plates and loaded with 3 µM fura 2-AM (Dojindo, Kumamoto, Japan) for 30 min. After cells were treated with the indicated reagents, fluorescence intensity (Excitation:340 or 380, Emission:510 nm) was measured by using a multimode microplate reader (Spark; TECAN, Switzerland) at 37 or 32°C.

## Statistical analysis

Data are expressed as mean ± SD. Differences between two groups were determined by Student's t-test. Differences between multiple group means were determined by two-way ANOVA combined with the Tukey's post hoc test. Differences between multiple groups with repeated measurements were evaluated by repeated one-way ANOVA or repeated two-way ANOVA combined with the post hoc test. Analyses were performed using GraphPad Prism 6 software (Graph Pad Software, La Jolla, CA, USA) or R version 4.0.2 (https://www.r-project.org). A p-value of <0.05 was considered statistically significant. Biological replicates indicate replicates of the same experiment conducted upon separately seeded culture on separate days. The number of biological replicates is described in the figure legend. For plate reader-based assay, n represents replicates that were acquired from different cells. In live-cell imaging assay, n represents replicates that were acquired from each cell through multiple set of experiments.

## Acknowledgements

This study was supported by the Japan Society for the Promotion of Science (JSPS) through the Grants-in-Aid for Scientific Research (C), (18K08112 and 21K08114, MT), Grants-in-Aid for Scientific Research on Innovative Areas (Thermal Biology) (16H01395. MT), the Agency for Medical Research and Development-Core Research for Evolutional Science and Technology (AMED-CREST) (MT), and the Ministry of Education, Culture, Sports, Science and Technology (MEXT)-supported program for Private University Research Branding Project (MT), and the Ichiro Kanehara Foundation (TKa). We are grateful to Dr. Kunitoshi Uchida and Dr. Jun Fujita for their valuable suggestions. We thank Naoko Sugaya, Masako Sakurai, and Rumiko Ochiai for their technical assistance.

## Additional information

### Funding

| Funder | Grant reference number | Author |
| --- | --- | --- |
| Japan Society for the Promotion of Science | 18K08112 | Masafumi Takahashi |
| Japan Society for the Promotion of Science | 21K08114 | Masafumi Takahashi |

| Funder | Grant reference number | Author |
|---|---|---|
| Japan Society for the Promotion of Science | 16H01395 | Masafumi Takahashi |
| Ichiro Kanehara Foundation for the Promotion of Medical Sciences and Medical Care | | Tadayoshi Karasawa |
| Japan Agency for Medical Research and Development | | Masafumi Takahashi |
| Ministry of Education, Culture, Sports, Science and Technology | | Masafumi Takahashi |

The funders had no role in study design, data collection and interpretation, or the decision to submit the work for publication.

## Author contributions

Tadayoshi Karasawa, Conceptualization, Funding acquisition, Investigation, Methodology, Project administration, Resources, Visualization, Writing - original draft, Writing – review and editing; Takanori Komada, Investigation, Supervision, Validation; Naoya Yamada, Chintogtokh Baatarjav, Investigation, Validation; Emi Aizawa, Investigation, Methodology, Resources; Yoshiko Mizushina, Validation; Sachiko Watanabe, Investigation, Methodology, Validation; Takayoshi Matsumura, Supervision, Writing – review and editing; Masafumi Takahashi, Conceptualization, Supervision, Writing – review and editing

## Author ORCIDs

Tadayoshi Karasawa http://orcid.org/0000-0002-6738-2360
Takanori Komada http://orcid.org/0000-0003-3360-3185
Yoshiko Mizushina http://orcid.org/0000-0002-9988-5755
Masafumi Takahashi http://orcid.org/0000-0003-2716-7532

## Decision letter and Author response

Decision letter https://doi.org/10.7554/eLife.75166.sa1
Author response https://doi.org/10.7554/eLife.75166.sa2

## Additional files

### Supplementary files

• Transparent reporting form

### Data availability

All data generated or analyzed during this study are included in the manuscript and supporting file; Source Data files have been provided for Figures 1-8.

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

# Appendix 1

## Appendix 1—key resources table

| Reagent type (species) or resource | Designation | Source or reference | Identifiers | Additional information |
|---|---|---|---|---|
| Cell line (*Homo sapiens*) | LentiX293T | Takara Bio | Z2180N | |
| Cell line (*H. sapiens*) | HeLa | Gift from Dr. Kenji Tago | N/A | |
| Cell line (*H. sapiens*) | *EF1-NLRP3-WT-mNeonGreen*-HeLa | This manuscript | N/A | Produced by lentiviral transduction |
| Cell line (*H. sapiens*) | *EF1-NLRP3-L353P-mNeonGreen*-HeLa | This manuscript | N/A | Produced by lentiviral transduction |
| Cell line (*H. sapiens*) | *EF1-NLRP3-D303N-mNeonGreen*-HeLa | This manuscript | N/A | Produced by lentiviral transduction |
| Cell line (*H. sapiens*) | *EF1-NLRP3-Y563N-mNeonGreen*-HeLa | This manuscript | N/A | Produced by lentiviral transduction |
| Cell line (*H. sapiens*) | *EF1-NLRP3-Y570C-mNeonGreen*-HeLa | This manuscript | N/A | Produced by lentiviral transduction |
| Cell line (*H. sapiens*) | *EF1-NLRP3-L353P-linker-mutant-mNeonGreen*-HeLa | This manuscript | N/A | Produced by lentiviral transduction |
| Cell line (*H. sapiens*) | *EF1-NLRP3-D303N-linker-mutant--mNeonGreen*-HeLa | This manuscript | N/A | Produced by lentiviral transduction |
| Cell line (*H. sapiens*) | *EF1-mNeonGreen*-HeLa | This manuscript | N/A | Produced by lentiviral transduction |
| Cell line (*H. sapiens*) | *EF1-ASC-GFP*-HeLa | This manuscript | N/A | Produced by lentiviral transduction |
| Cell line (*H. sapiens*) | THP-1 | ATCC | TIB-202 | |
| Cell line (*H. sapiens*) | *ASC* KO THP-1 | *Aizawa et al., 2020* | N/A | Produced by lentiviral transduction |
| Cell line (*H. sapiens*) | *EF1-NLRP3-WT-mNeonGreen/ASC* KO THP-1 | This manuscript | N/A | Produced by lentiviral transduction |
| Cell line (*H. sapiens*) | *EF1-NLRP3-L353P-mNeonGreen/ASC* KO THP-1 | This manuscript | N/A | Produced by lentiviral transduction |
| Cell line (*H. sapiens*) | *EF1-NLRP3-D303N-mNeonGreen/ASC* KO THP-1 | This manuscript | N/A | Produced by lentiviral transduction |
| Cell line (*H. sapiens*) | *TRE-NLRP3-WT*-THP-1 | This manuscript | N/A | Produced by lentiviral transduction and limiting dilution |
| Cell line (*H. sapiens*) | *TRE-NLRP3-L353P*-THP-1 | This manuscript | N/A | Produced by lentiviral transduction and limiting dilution |
| Cell line (*H. sapiens*) | *TRE-NLRP3-L353P/ASC* KO THP-1 | This manuscript | N/A | Produced by lentiviral transduction |
| Cell line (*H. sapiens*) | *TRE-NLRP3-L353P/NEK7* KO THP-1 | This manuscript | N/A | Produced by lentiviral transduction |

*Appendix 1 Continued on next page*

*Appendix 1 Continued*

| Reagent type (species) or resource | Designation | Source or reference | Identifiers | Additional information |
|---|---|---|---|---|
| Cell line (*H. sapiens*) | *TRE-NLRP3-L353P/CASP1* KO THP-1 | This manuscript | N/A | Produced by lentiviral transduction |
| Cell line (*H. sapiens*) | *TRE-NLRP3-D303N*-THP-1 | *Aizawa et al., 2020* | N/A | Produced by lentiviral transduction and limiting dilution |
| Cell line (*H. sapiens*) | *TRE-NLRP3-D303N/ASC* KO THP-1 | This manuscript | N/A | Produced by lentiviral transduction |
| Cell line (*H. sapiens*) | *EF1-ASC-GFP/TRE-NLRP3-L353P*-THP-1 | This manuscript | N/A | Produced by lentiviral transduction |
| Cell line (*H. sapiens*) | *EF1-ASC-BFP/TRE-NLRP3-L353P*-THP-1 | This manuscript | N/A | Produced by lentiviral transduction |
| Cell line (*H. sapiens*) | *EF1-ASC-BFP/TRE-NLRP3-WT-mNeonGreen*-THP-1 | This manuscript | N/A | Produced by lentiviral transduction |
| Sequence-based reagent | sgRNA sequence for GFP | *Aizawa et al., 2020* | N/A | GAGCTGGACG GCGACGTAAA |
| Sequence-based reagent | sgRNA sequence for apoptosis-associated speck-like protein containing a caspase recruitment domain (ASC) | *Aizawa et al., 2020* | N/A | CAGCACGTTA GCGGTGAGCT |
| Sequence-based reagent | sgRNA sequence for NEK7#1 | This manuscript | N/A | ATTACAGAAG GCCTTACGAC |
| Sequence-based reagent | sgRNA sequence for NEK7#2 | This manuscript | N/A | ATAGCCCATA TCCGGTCGTA |
| Sequence-based reagent | sgRNA sequence for CASP1 | *Aizawa et al., 2020* | N/A | AAGCTGTTTA TCCGTTCCAT |
| Recombinant DNA reagent | CSCAMCS | RIKEN BRC | RDB05963 | Lentiviral vector for stable gene expression |
| Recombinant DNA reagent | CSIV-TRE-RfA-CMV-KT | RIKEN BRC | RDB12876 | Lentiviral vector for inducible gene expression |
| Recombinant DNA reagent | LentiCRISPRv2 | Addgene | #52,961 | Lentiviral vector expressing sgRNA |
| Transfected construct (*H. sapiens*) | CSEF1-NLRP3-WT-mNeonGreen | This manuscript | N/A | Lentiviral vector for stable gene expression |
| Transfected construct (*H. sapiens*) | CSEF1-NLRP3-L353P-mNeonGreen | This manuscript | N/A | Lentiviral vector for stable gene expression |
| Transfected construct (*H. sapiens*) | CSEF1-NLRP3-D303N-mNeonGreen | This manuscript | N/A | Lentiviral vector for stable gene expression |
| Transfected construct (*H. sapiens*) | CSEF1-NLRP3-Y563N-mNeonGreen | This manuscript | N/A | Lentiviral vector for stable gene expression |
| Transfected construct (*H. sapiens*) | CSEF1-NLRP3-Y570C-mNeonGreen | This manuscript | N/A | Lentiviral vector for stable gene expression |
| Transfected construct (*H. sapiens*) | CSEF1-NLRP3-L353P-linker-mutant-mNeonGreen | This manuscript | N/A | Lentiviral vector for stable gene expression |
| Transfected construct (*H. sapiens*) | CSEF1-NLRP3-D303N-linker-mutant -mNeonGreen | This manuscript | N/A | Lentiviral vector for stable gene expression |

*Appendix 1 Continued on next page*

*Appendix 1 Continued*

| Reagent type (species) or resource | Designation | Source or reference | Identifiers | Additional information |
|---|---|---|---|---|
| Transfected construct (*H. sapiens*) | CSEF1-mNeonGreen | This manuscript | N/A | Lentiviral vector for stable gene expression |
| Transfected construct (*H. sapiens*) | CSEF1-ASC-GFP | This manuscript | N/A | Lentiviral vector for stable gene expression |
| Transfected construct (*H. sapiens*) | CSEF1-ASC-BFP | This manuscript | N/A | Lentiviral vector for stable gene expression |
| Transfected construct (*H. sapiens*) | CSIV-TRE-NLRP3-WT-CMV-KT | This manuscript | N/A | Lentiviral vector for inducible gene expression |
| Transfected construct (*H. sapiens*) | CSIV-TRE-NLRP3-L353P-CMV-KT | This manuscript | N/A | Lentiviral vector for inducible gene expression |
| Transfected construct (*H. sapiens*) | CSIV-TRE-NLRP3-D303N-CMV-KT | *Aizawa et al., 2020* | N/A | Lentiviral vector for inducible gene expression |
| Transfected construct (*H. sapiens*) | CSIV-TRE-NLRP3-WT-mNeonGreen-CMV-KT | This manuscript | N/A | Lentiviral vector for inducible gene expression |
| Transfected construct (*H. sapiens*) | CSIV-TRE-NLRP3-L353P-mNeonGreen -CMV-KT | This manuscript | N/A | Lentiviral vector for inducible gene expression |
| Transfected construct (*H. sapiens*) | LentiCRISPRv2 sgGFP | *Aizawa et al., 2020* | N/A | Lentiviral vector expressing sgRNA |
| Transfected construct (*H. sapiens*) | LentiCRISPRv2 sgASC | *Aizawa et al., 2020* | N/A | Lentiviral vector expressing sgRNA |
| Transfected construct (*H. sapiens*) | LentiCRISPRv2 sgNEK7 | This manuscript | N/A | Lentiviral vector expressing sgRNA |
| Transfected construct (*H. sapiens*) | LentiCRISPRv2 sgCASP1 | *Aizawa et al., 2020* | N/A | Lentiviral vector expressing sgRNA |
| Antibody | Rabbit polyclonal anti-NEK7 | Abcam | EPR4900 | WB (1:1000) |
| Antibody | Mouse monoclonal anti-NLRP3 | Adipogen | AG-20B-0014 | WB (1:2000) |
| Antibody | Rabbit polyclonal anti-ASC | Adipogen | AG-25B-0006 | WB (1:2000) |
| Antibody | Rabbit monoclonal anti-caspase-1 (D7F10) | Cell Signaling Technology | #3,866 | WB (1:1000) |
| Antibody | Rabbit polyclonal anti-IL-1β | Santa Cruz | sc-7884 | WB (1:1000) |
| Antibody | Mouse monoclonal anti-β-actin | Sigma-Aldrich | A5441 | WB (1:4000) |
| Commercial assay or kit | Lentiviral qPCR Titration Kit | Applied Biological Materials | #LV900 | |
| Commercial assay or kit | Human IL-1 beta/IL-1F2 DuoSet ELISA | R&D Systems | DY201 | |
| Commercial assay or kit | Super Script VILO cDNA Synthesis kit | Thermo Fisher Scientific | | |
| Chemical compound, drug | MCC950 | AdipoGen | AG-CR1-3615-M005 | |
| Chemical compound, drug | DRAQ5 | Biolegend | 424,101 | 1 µM |
| Chemical compound, drug | Probenecid | Cayman | 14,981 | |
| Chemical compound, drug | Trovafloxacin mesylate | Cayman | 9000303 | |

*Appendix 1 Continued on next page*

*Appendix 1 Continued*

| Reagent type (species) or resource | Designation | Source or reference | Identifiers | Additional information |
|---|---|---|---|---|
| Chemical compound, drug | 4',6-diamidino-2-phenylindole, dihydrochloride | DOJINDO | D523 | 1 µg/mL |
| Chemical compound, drug | Fura-2/AM | DOJINDO | F015 | 3 µM |
| Chemical compound, drug | Hoechst33342 | DOJINDO | H342 | 1 µg/mL |
| Chemical compound, drug | PEI MAX | Polyscience | 24765–1 | |
| Chemical compound, drug | VX-765 | Selleck | S2228 | |
| Chemical compound, drug | Pluronic F127 | Sigma-Aldrich | P2443 | 0.04% |
| Chemical compound, drug | Puromycin | Sigma-Aldrich | P8833 | 2 µg/mL |
| Chemical compound, drug | Fluo-8 | Santa Cruz | Sc-362562 | 4 µM |
| Chemical compound, drug | Bis(sulfosuccinimidyl)suberate | Thermo Fisher Scientific | 21,580 | 2 mM |
| Chemical compound, drug | SYTOX Deep Red | Thermo Fisher Scientific | S11380 | 100 nM |
| Chemical compound, drug | Doxycycline Hydrochloride n-Hydrate | Wako | 049–31121 | |
| Chemical compound, drug | Phorbol 12-Myristate 13-Acetate | Wako | 162–23591 | 200 nM |
| Software, algorithm | GraphPad Prism 6 | Graph Pad Software | RRID: SCR_002798 | |
| Software, algorithm | ImageJ/FIJI (2.1.0/1.53 c) | *Schindelin et al., 2012* | RRID: SCR_002285 | |
| Software, algorithm | IUpred2A | *Mészáros et al., 2018* | RRID: SCR_014632 | |
| Software, algorithm | R version 4.0.2 | R project | RRID: SCR_001905 | |

