## [Editor Report]

Gain of function mutations in NLRP3 are associated with a group of autoinflammatory diseases called the cryopyrin-associated periodic syndromes (CAPS). Karasawa and colleagues reveal that CAPS-associated NLRP3 mutants form cryo-sensitive aggregates that promote NLRP3 inflammasome assembly and Caspase-1 activation through elegant immunofluorescence studies, providing mechanistic insights into CAPS.

---

## [Decision Letter]

**Decision letter after peer review:**

Thank you for submitting your article "Cryo-sensitive aggregation triggers NLRP3 inflammasome assembly in cryopyrin-associated periodic syndrome" for consideration by *eLife*. Your article has been reviewed by 3 peer reviewers, one of whom is a member of our Board of Reviewing Editors, and the evaluation has been overseen by Carla Rothlin as the Senior Editor. The following individuals involved in review of your submission have agreed to reveal their identity: Seth L Masters (Reviewer #2); Hal Hoffman (Reviewer #3).

Essential revisions:

1. The role of Ca in constitutive aggregation of NLRP3 CAPS mutants and the subsequent inflammasome assembly, and role for caspase 1 should be clarified as suggested by Reviewer 1.

2. Additional triggers of NLRP3 should be tested for the reasons described by Reviewer 2.

3. Whether the cryo-sensitive aggregation a common characteristic of other autoinflammation-associated NLRP3 mutants should be examined.

4. Attempt to address why a CINCA mutation seen in patients in which cold is not a trigger of clinical symptoms have similar cold induced aggregation.

*Reviewer #1 (Recommendations for the authors):*

It is not clear if Ca is required for both the constitutive aggregation of NLRP3 CAPS mutants and the subsequent inflammasome assembly. Is the aggregation of NLRP3 CAPS mutants the trigger for Ca influx? This would then argue against a role for Ca in the aggregation of NLRP3 CAPS mutants.

Casp1 inhibitor impairing Ca influx suggests that Ca influx amplifies rather than triggers the activation of NLRP3 CAPS mutants. In which case, Ca-independent aggregation of NLRP3 CAPS mutants should be demonstrated. Furthermore, the authors should address how Ca influx amplifies the activation of NLRP3 CAPS mutants.

Is this cryo-sensitive aggregation a common characteristic of other autoinflammation-associated NLRP3 mutants?

Biochemical demonstration of aggregation of NLRP3 CAPS mutants would strengthen the immunofluorescence data in Figure 1 (L353P and D303N).

Figure 3 and its supplement should include direct comparisons of WT NLRP3 and NLRP3 mutants.

NEK7 deletion should be verified by western blotting.

*Reviewer #2 (Recommendations for the authors):*

Can the authors quantify the area of mutant NLRP3 foci per cell, at normal and low temperatures. Presumably this is important as the aggregate size (and number) are likely to be functions of temperature?

Again as suggested above, consider additional triggers for NLRP3 activation in case temperature affects nigericin and silica in independent ways. Perhaps a more direct trigger, things that disturb the TGN, what about K^+^ efflux independent activation with transfected imiquimod?

In general, only two mutations in NLRP3 are considered, when there are a wider variety associated with FCAS. Some of these might have alternate mechanisms of activation, be further from the ATP binding site or oligomerise the molecule in other ways, so it would be good to confirm that a variety still form temperature dependent foci. It may also be informative to make a mutant of the polybasic region to establish if this is involved in mutant NLRP3 foci formation, however presumably these do not colocalise with the dTGN or another structure within the cell?

It is very confusing that the increase in specks for mutant NLRP3 does not occur with Casp1 inhibitors as this does not seem consistent with increased mutant NLRP3 foci observed in the absence of ASC. Perhaps mutant NLRP3 foci need to be quantified with the Caspase-1 inhibitors, or some other explanation to this apparent contradiction? In general I remain unclear about the importance of feedforward effects for Caspase-1 in this process and would consider to remove this data unless the physiological significance is clear. At the very least, more controls like the genetic deletion of Caspase-1 would be critical to establish that this is not a non-specific effect of VX-765.

*Reviewer #3 (Recommendations for the authors):*

Abstract

Concise and clear.

Introduction

Initial symptoms described should be described as unique or differentiating clinical features of the subtypes.

Cryopyrin refers to the protein, not the gene. Initial identification of NLRP3 was in 2001.

Methods

Adequate detail of techniques

Results

Overall mostly clear presentation of data in main and supplemental figures and mostly accurate interpretation.

I found the insertion of figures into the text more difficult to read. While the figure legends are accurate the authors should label the subfigures better since it is difficult to keep track of what is transfected in each sub figure.

The use of two mutant constructs is a major strength, but there are some clear differences between the two and the authors only study both constructs in some of the experiments which may ignore some of these differences.

Specific comments

Figure 1A – I don't understand the difference between foci and speck, especially since they are both shown to be aggregates later in figure 2. In the images shown it appears than the FCAS transfected cells have several low intensity areas and the CINCA transfected cells have similar low intensity areas but also have one speck similar to the ASC GFP transfected cells.

Figure 1D-E. This is the most novel finding of the paper in that mutants have increased foci number while WT have reduced or unchanged foci number when cultured at 32 C. However, the cold induced data is not consistent with clinical findings since CINCA patients do not experience cold induced symptoms. The authors should at least comment on this and make an attempt to explain it.

One of the most exciting findings of this figure is that the data is consistent with the severity of the disease associated with the specific mutation. These assays could provide quantitative methods to quantify mutation pathogenicity.

Figure 2

Clear demonstration of aggregates and not LLPS.

However, if they are both aggregates, then why are the authors describing them differently.

Other than the LLPS inhibitor, I don't understand what is different about supplemental Figure 2.

Figure 3 and supplemental Figure 3

The use of the TET ON system is novel, and clearly shows increased NLRP3 expression and dimerization as well as ASC solublilty changes when induced and with temperature but I don't understand why it proves that the tag isn't responsible for the aggregate formation.

I found the reduction of cold induced IL-1B release with extrinsic stimulation of WT cells to be interesting. This should be emphasized as more evidence of mutation specific effects.

This figure might actually help the authors in that the cold induction of IL-1B expression and release in the CINCA transfected cells is much more modest than in the FCAS transfected cells. Its possible that CINCA patient do have cold induction but don't notice it since their baseline inflammation is so high and there is so little change with cold.

Do foci number correlate well with IL-1B release?

Figure 4

Clear.

This is some of the strongest data of the paper.

Potassium and NEK7 data is interesting and could be moved from supplemental.

It might be useful to see if the CINCA mutation has some K and NEK7 dependence.

Figure 5

Clear and novel.

Are results the same for CINCA mutations.

Figure 6

Clear and translational.

Involvement of Caspase 1 in calcium mobilization is novel finding.

6B I don't understand.

6D Interesting that casp 1 inhibition affects ASC solubility.

MCC950 data could be moved from Supplemental to main.

Are results the same for CINCA mutations.

Figure 7

Clear and helpful.

Discussion

Concisely written.

Data supports conclusions.

Limitations discussed.

Need to discuss differences in mutation and try to relate to clinical findings.

References – Mostly Adequate.

---

## [Author Response]

Essential revisions:1. The role of Ca in constitutive aggregation of NLRP3 CAPS mutants and the subsequent inflammasome assembly, and role for caspase 1 should be clarified as suggested by Reviewer 1.

Thank you for pointing out this issue. We showed that ca^2+^ promotes both the initial aggregation and subsequent inflammasome assembly. Although the presence of ca^2+^ enhanced the aggregation of mutated NLRP3, the aggregation still occurred in the absence of ca^2+^ (Figure 5C­–E, Figure 5­—figure supplement 2A–C). The initial aggregation of mutated NLRP3 triggered ca^2+^ influx because inflammasome assembly visualized by ASC-BFP occurred prior to ca^2+^ influx (Figure 6G–I). Furthermore, ca^2+^ influx was mediated by caspase-1 because pharmacological inhibition of caspase-1 prevented ca^2+^ influx and ASC speck formation induced by mutated NLRP3 (Figure 7A–E). In the revised manuscript, the involvement of caspase-1 was further validated by genetically mutated cells (Figure 7G and H). With regard to the mechanism of ca^2+^ influx, we found that inhibition of pannexin 1 attenuated ca^2+^ influx and the formation of inflammasome assembly (Figure 8A–H). We have added the data and discussed these issues in the revised manuscript (page 8, paragraph 1, page 9, paragraph 1 and 2, and page 12, paragraph 2).

2. Additional triggers of NLRP3 should be tested for the reasons described by Reviewer 2.

According to the suggestions, we investigated additional regulatory mechanisms of inflammasome activation induced by mutated NLRP3. Recent studies have suggested that dispersion of TGN plays a critical role in both K^+^ efflux-dependent and independent inflammasome activation (Nature, 2018;564:71). Since the polybasic linker in NLRP3 is required for its TGN localization, we introduced mutation in the polybasic linker of mutated NLRP3. However, the NLRP3 polybasic linker mutants formed cryo-sensitive foci, indicating that TGN localization is dispensable for inflammasome activation induced by mutated NLRP3 (Figure 4C–E and Figure 4—figure supplement 1F). Instead, according to the reviewer’s suggestion, we assessed the effect of pannexin 1 inhibitor and found that pannexin 1 was involved in ca^2+^ influx and subsequent inflammasome assembly induced by FCAS-associated NLRP3 mutant (Figure 8A–H and Figure 8—figure supplement 1A–D). We have added the data and mentioned these issues in the revised manuscript (page 7, paragraph 1, and page 9, paragraph 2).

3. Whether the cryo-sensitive aggregation a common characteristic of other autoinflammation-associated NLRP3 mutants should be examined.

We appreciate this important comment. In addition to L353P and D303N mutants, we developed FCAS-associated Y563N and CINCA-associated Y570C mutants. Whereas previously developed L353P and D303N mutants are located in NBD, these two mutations are located in NLRC4HD domain of NLRP3 (Figure 1—figure supplement 1A). Expectedly, both two mutants form cryo-sensitive aggregates (Figure 1I and Figure 1—figure supplement 1B–E). We have added the data and mentioned these issues in the revised manuscript (page 4, paragraph 3, and page 5, paragraph 1).

4. Attempt to address why a CINCA mutation seen in patients in which cold is not a trigger of clinical symptoms have similar cold induced aggregation.

We appreciate this invaluable comment. We further analyzed FCAS-associated L353P and Y563N mutants and CINCA-associated D303N and Y570C mutants. Although all of the analyzed mutated NLRP3 exhibited cryo-sensitivity, the sensitivity of FCAS mutants was more prominent than that of CINCA mutants (Figure 1F and I, Figure 1—figure supplement 1D–G). Moreover, the foci number of CINCA-associated mutants was higher than that of FCAS-associated mutants at 37ºC. Thus, it is likely that CINCA-associated NLRP3 mutants exhibit cryo-sensitivity, but their ability to form aggregates is potent enough to induce inflammasome activation at 37ºC. We have added the data and discussed these issues in the revised manuscript (page 5, paragraph 1, and page 11, paragraph 1).

Reviewer #1 (Recommendations for the authors):It is not clear if Ca is required for both the constitutive aggregation of NLRP3 CAPS mutants and the subsequent inflammasome assembly. Is the aggregation of NLRP3 CAPS mutants the trigger for Ca influx? This would then argue against a role for Ca in the aggregation of NLRP3 CAPS mutants.

(Please also see the response to the Editor #1):

Thank you for pointing out this important issue. Ca^2+^ promotes both the initial aggregation and subsequent inflammasome assembly. However, aggregation of mutated NLRP3 occurred in the absence of ca^2+^. NLRP3 L353P mutant and D303N mutant formed cold-exposure-induced foci in ca^2+^-depleted condition (Figure 5­—figure supplement 2A–C). As the reviewer suggested, the initial aggregation of mutated NLRP3 triggered ca^2+^ influx because inflammasome assembly visualized by ASC-BFP occurred prior to ca^2+^ influx (Figure 6G–I). With regard to the mechanism of ca^2+^ influx, we found that inhibition of pannexin 1 attenuated ca^2+^ influx and inflammasome assembly (Figure 8A–H). We have added the data and discussed these issues in the revised manuscript (page 9, paragraph 2, and page 12, paragraph 2).

Casp1 inhibitor impairing Ca influx suggests that Ca influx amplifies rather than triggers the activation of NLRP3 CAPS mutants. In which case, Ca-independent aggregation of NLRP3 CAPS mutants should be demonstrated. Furthermore, the authors should address how Ca influx amplifies the activation of NLRP3 CAPS mutants.

We appreciate your constructive comment. We agree that ca^2+^ influx functions as an amplifier of inflammasome assembly induced by mutated NLRP3. As the reviewer suggested, cold exposure promoted aggregation of mutated NLRP3 even in the absence of ca^2+^ (Figure 5­—figure supplement 2A–C). However, the supplementation of ca^2+^ promoted aggregation of mutated NLRP3 (Figure 5E). Previous studies have suggested that cationic ions regulate protein aggregates such as α-synuclein (Sci Rep. 2018;8:1895, doi: 10.1038/s41598-018-20320-5.). Therefore, we speculate that the influx of ca^2+^ is a direct regulator of mutated NLRP3 aggregation. We have added the data and mentioned these issues in the revised manuscript (page 8, paragraph 1).

Is this cryo-sensitive aggregation a common characteristic of other autoinflammation-associated NLRP3 mutants?

(Please also see the response to the Editor #3):

We appreciate this invaluable comment. According to the comment, we developed the cells expressing other CAPS-associated NLRP3 mutants: FCAS-associated Y563N and CINCA-associated Y570C. Expectedly, both two mutants form cryo-sensitive aggregate (Figure 1I, Figure1—figure supplement 1B–G). Moreover, sensitivity to cold exposure is more prominent in FCAS-associated Y563N than CINCA-associated Y570C. We have added the data and mentioned these issues in the revised manuscript (page 5, paragraph 1).

Biochemical demonstration of aggregation of NLRP3 CAPS mutants would strengthen the immunofluorescence data in Figure 1 (L353P and D303N).

Thank you for this valuable suggestion. To assess the aggregation of mutated NLRP3 without being affected by fluorescent tag, we developed the *ASC-KO /TRE-NLRP3-L353P*- and *ASC-KO /TRE-NLRP3-D303N*-THP-1 cells and showed that the mutated NLRP3 were detected in the detergent-insoluble fraction in the absence of ASC. This biochemical experiment supports that mutated NLRP3 forms aggregates (Figure 3—figure supplement 1C). We have added the data and mentioned these issues in the revised manuscript (page 6, paragraph 1).

Figure 3 and its supplement should include direct comparisons of WT NLRP3 and NLRP3 mutants.

Thank you for your important comment. We agree that direct comparisons between mutants are helpful to validate the potency of each mutant. In our experiment, however, transduction of lentiviral vector encoding mutated NLRP3 induced massive cell death in the presence of ASC probably due to carrying over of mutated NLRP3 protein produced in 293T cells during lentiviral production. Therefore, we established each cell line from survived cells with limiting dilution. IL-1β release and ASC speck formation were evaluated in each clone. Instead, we directly compared the effect of WT and L353P on ASC recruitment to the aggregates (Figure 3A–C). We would like to validate the potency to produce IL-1β in each mutant in future work.

NEK7 deletion should be verified by western blotting.

According to this suggestion, we confirmed the deficiency of NEK7 by western blotting (Figure 4—figure supplement 1C). We have added the data and mentioned this issue in the revised manuscript (page 7, paragraph 1).

Reviewer #2 (Recommendations for the authors):As suggested above, can the authors quantify the area of mutant NLRP3 foci per cell, at normal and low temperatures. Presumably this is important as the aggregate size (and number) are likely to be functions of temperature?

Thank you for this important comment. We quantified the area and number of foci formed by NLRP3 mutants. Although the total area of foci was increased by cold exposure, the mean area of foci was not increased by cold exposure (Figure 1—figure supplement 1F and G). In contrast, the number of foci was increased by cold exposure (Figure 1H and I). These results suggest that the temperature affects the number of foci rather than their size. We have added the data and mentioned these issues in the revised manuscript (page 5, paragraph 1).

Again as suggested above, consider additional triggers for NLRP3 activation in case temperature affects nigericin and silica in independent ways. Perhaps a more direct trigger, things that disturb the TGN, what about K^+^ efflux independent activation with transfected imiquimod?

Thank you for this helpful comment. A previous report has suggested that imiquimod induces TGN-dispersion and inflammasome activation in a K^+^ efflux-independent manner. In our experiment with THP-1 cells, however, the treatment with imiquimod failed to induce IL-1β release (Author response image 1). Therefore, we were not able to evaluate the effect of cold exposure on imiquimod-induced response. Instead, we investigated the involvement of TGN because the dispersion of TGN plays a critical role in both K^+^-efflux-dependent and independent inflammasome activation. Although we introduced mutation into polybasic linker, which interacts with phosphorylated phosphatidylinositide on TGN membrane, the NLRP3 polybasic linker mutant forms cryo-sensitive aggregation (Figure 4C–E). On the other hand, we found that pannexin 1 is involved in ca^2+^ influx and subsequent inflammasome assembly induced by NLRP3 L353P mutant (Figure 8A–H). We have added the data and mentioned these issues in the revised manuscript (page 7, paragraph 1, and page 9, paragraph 2).

**Author response image 1. sa2fig1:** The effect of imiquimod on IL-1β release in THP-1 cells. THP-1 cells were differentiated with PMA for 24 h and then treated with nigericin, imiquimod, or nanosilica at 37ºC or 32ºC for 6 h. The levels of IL-1β in the supernatants were assessed by ELISA (n = 3). Data are expressed as the mean ± SD.

In general, only two mutations in NLRP3 are considered, when there are a wider variety associated with FCAS. Some of these might have alternate mechanisms of activation, be further from the ATP binding site or oligomerise the molecule in other ways, so it would be good to confirm that a variety still form temperature dependent foci.

(Please also see the response to the Editor #3):

We appreciate this helpful comment. According to the comment, we developed the cells expressing other CAPS-associated NLRP3 mutants: FCAS-associated Y563N and CINCA-associated Y570C. These two mutations are located in NLRC4HD domain of NLRP3. Expectedly, both two mutants form cryo-sensitive aggregates (Figure 1I, Figure 1—figure supplement 1B–G). We have added the data and mentioned these issues in the revised manuscript (page 4, paragraph 3, and page 5, paragraph 1).

It may also be informative to make a mutant of the polybasic region to establish if this is involved in mutant NLRP3 foci formation, however presumably these do not colocalise with the dTGN or another structure within the cell?

We appreciate this valuable suggestion. As described in the response to the comment #2, we developed the mutants lacking three lysine residues (K129, K131, and K132) in the polybasic linker of NLRP3 by introducing alanine. Notably, the polybasic region mutant still formed foci (Figure 4C). Although foci formation by D303N mutant was decreased by polybasic linker mutation, cold exposure-enhanced foci formation was detected even in the polybasic linker mutant (Figure 4D and E). These results suggest that localization to TGN is dispensable for cryo-sensitive aggregation of mutated NLRP3. We have added the data and mentioned these issues in the revised manuscript (page 7, paragraph 1).

It is very confusing that the increase in specks for mutant NLRP3 does not occur with Casp1 inhibitors as this does not seem consistent with increased mutant NLRP3 foci observed in the absence of ASC. Perhaps mutant NLRP3 foci need to be quantified with the Caspase-1 inhibitors, or some other explanation to this apparent contradiction? In general I remain unclear about the importance of feedforward effects for Caspase-1 in this process and would consider to remove this data unless the physiological significance is clear. At the very least, more controls like the genetic deletion of Caspase-1 would be critical to establish that this is not a non-specific effect of VX-765.

Thank you for this thoughtful comment. The inhibition of caspase-1 with VX-765 partially attenuated inflammasome assembly (Figure 7D and E). Therefore, we postulate that caspase-1-mediated ca^2+^ influx was dispensable for the initial aggregation of NLRP3 mutants. According to the suggestions, we further validate the effect of *CASP1*-deficiency on mutated NLRP3-mediated ca^2+^ influx and inflammasome assembly. Indeed, *CASP1*-deficiency attenuated ca^2+^ influx and ASC speck formation induced by mutated NLRP3 (Figure 7G and H). We have added the data and mentioned these issues in the revised manuscript, (page 9, paragraph 1).

Reviewer #3 (Recommendations for the authors):IntroductionInitial symptoms described should be described as unique or differentiating clinical features of the subtypes.

Thank you for this helpful suggestion. We described symptoms of each of the three syndromes in detail as follows: “FCAS is the mildest form of CAPS and is characterized by cold-induced fever, arthralgia, urticaria, and conjunctivitis. MWS is accompanied by systemic amyloidosis and progressive hearing loss. CINCA is the most severe phenotype and is characterized by central nervous system inflammation, bone deformities, and chronic conjunctivitis.” (page 3, paragraph 1).

Cryopyrin refers to the protein, not the gene. Initial identification of NLRP3 was in 2001.

We do apologize for our mistake. According to the suggestion, we have amended the sentence as follows:

“Genetic causes of these disorders are gain-of-function mutations in the Nucleotide-binding oligomerization domain, leucine-rich repeat and pyrin domain containing 3 (NLRP3) gene, encoding cryopyrin (Hoffman et al., 2001; Brydges et al., 2009; Kuemmerle-Deschner, 2015).” (page 3, paragraph 3).

ResultsOverall mostly clear presentation of data in main and supplemental figures and mostly accurate interpretation.I found the insertion of figures into the text more difficult to read. While the figure legends are accurate the authors should label the subfigures better since it is difficult to keep track of what is transfected in each sub figure.

We do apologize for troubling you. According to the suggestion, we have amended the arrangement of figures. In the revised manuscript, figures are provided separately. Further, we have added the label of the transduced gene in each subfigure.

The use of two mutant constructs is a major strength, but there are some clear differences between the two and the authors only study both constructs in some of the experiments which may ignore some of these differences.

Thank you for this invaluable comment. In the previous manuscript, we analyzed FCAS-associated L353P mutant and CINCA-associated D303N mutant. In some experiments, we mainly analyzed L353P mutant to clarify the mechanisms of its cryo-sensitivity. In the revised manuscript, we performed additional experiments using D303N mutant and confirmed that D303N mutant-mediated inflammasome assembly is independent of NEK7 and dispensable for TGN (Figure 4—figure supplement 1E, and Figure 4D). Meanwhile, although the expression of D303N mutant also caused ca^2+^ influx (Figure 6—figure supplement 1A and B), the inhibition of pannexin 1 failed to inhibit IL-1β release induced by D303N mutant (Figure 8—figure supplement 1G). D303N mutant was less sensitive to ca^2+^ because ca^2+^-depletion partially inhibited IL-1β release induced by D303N (Figure 5—figure supplement 1F and Figure 5—figure supplement 2B). Moreover, a remarkable difference between FCAS and CINCA-associated mutants was observed in the formation of aggregates at 37ºC. Whereas foci formation in FCAS-mutants was markedly induced at 32ºC, a large number of foci were detected even at 37ºC in CINCA-associated mutants (Figure 1I). These results suggest that D303N mutant displays constitutive active properties. We have added the data and discussed these issues in the revised manuscript (page 11, paragraph 1, and page 12, paragraph 2).

Specific commentsFigure 1A – I don't understand the difference between foci and speck, especially since they are both shown to be aggregates later in figure 2. In the images shown it appears than the FCAS transfected cells have several low intensity areas and the CINCA transfected cells have similar low intensity areas but also have one speck similar to the ASC GFP transfected cells.

Thank you for pointing out this issue. Previous studies have suggested that ASC forms a large speck aggregate (J Biol Chem. 1999;274, 33835. doi: 10.1074/jbc.274.48.33835.). ASC generally forms a single speck per cell, and the size of the speck is 1–2 µm in a diameter (Cell death Differ. 2007. 2007;14:1590, doi: 10.1038/sj.cdd.4402194.). In contrast, NLRP3 mutants form small multiple foci in the cells. To validate the number of mutated NLRP3 foci per cell, we analyzed the data of high content analysis and confirmed that mutated NLRP3 forms one to multiple foci in the cells (Figure 1—figure supplement 1D).

Figure 1D-E. This is the most novel finding of the paper in that mutants have increased foci number while WT have reduced or unchanged foci number when cultured at 32 C. However, the cold induced data is not consistent with clinical findings since CINCA patients do not experience cold induced symptoms. The authors should at least comment on this and make an attempt to explain it.

(Please also see response to the Editor #3):

We appreciate this important comment. We further analyzed FCAS-associated Y563N mutant and CINCA-associated Y570C mutant in the revised manuscript. Although the analyzed mutated NLRP3 exhibited cryo-sensitivity, the sensitivity of FCAS mutants was more prominent than that of CINCA (Figure 1E–I and Figure 1—figure supplement 1E and F). Moreover, foci number of CINCA-associated mutants was higher than that of FCAS-associated mutants at 37ºC. Thus, we postulate that CINCA-associated NLRP3 mutants exhibit cryo-sensitivity, but their ability to form aggregates is potent enough to induce inflammasome activation at 37ºC. We have added the data and discussed these issues in the revised manuscript (page 11, paragraph 1).

One of the most exciting findings of this figure is that the data is consistent with the severity of the disease associated with the specific mutation. These assays could provide quantitative methods to quantify mutation pathogenicity.

Thank you for your positive comment. Because we evaluated only four mutants in this study, it is difficult to determine the severity of CAPS associated with specific mutations by the number of mutated NLRP3 foci. Therefore, we would like to investigate the association between foci number and disease severity in future work.

Other than the LLPS inhibitor, I don't understand what is different about supplemental Figure 2.

I’m sorry for our insufficient explanation. In figure-supplement 2, we analyzed partially bleached D303N aggregates and wholly bleached L353P aggregates, separately (Figure 2—figure supplement 2A and C), whereas partially bleached L353P aggregate were analyzed in figure 2A and B. In addition, as a positive control of FRAP, we have added the data on FRAP analysis of unfused mNeonGreen protein (Figure 2—figure supplement 2E and F). We have added the data and mentioned these issues in the revised manuscript (page 5, paragraph 2).

Figure 3 and supplemental Figure 3The use of the TET ON system is novel, and clearly shows increased NLRP3 expression and dimerization as well as ASC solublilty changes when induced and with temperature but I don't understand why it proves that the tag isn't responsible for the aggregate formation.

We do apologize for our insufficient explanation. We developed and used the cells expressing full length NLRP3 without any tags or fluorescent proteins under the TRE promoter in these experiments. The induction of mutated NLRP3 promoted ASC oligomerization and ASC speck formation (Figure 3D, E, and G). Furthermore, to investigate the aggregation of mutated NLRP3 without being affected by fluorescent tag, we developed the *ASC-KO /TRE-NLRP3-L353P*- and *ASC-KO /TRE-NLRP3-D303N*-THP-1 cells. The mutated NLRP3 proteins were detected in the detergent-insoluble fraction in the absence of ASC, supporting that mutated NLRP3 forms aggregates (Figure 3—figure supplement 1D). We have added the data and mentioned these issues in the revised manuscript (page 6, paragraph 1).

I found the reduction of cold induced IL-1B release with extrinsic stimulation of WT cells to be interesting. This should be emphasized as more evidence of mutation specific effects.

According to the suggestion, we have emphasized and discussed this issue in the revised manuscript (page 11, paragraph 1).

This figure might actually help the authors in that the cold induction of IL-1B expression and release in the CINCA transfected cells is much more modest than in the FCAS transfected cells. Its possible that CINCA patient do have cold induction but don't notice it since their baseline inflammation is so high and there is so little change with cold.

Thank you for your comment. As the reviewer mentioned, we postulate that CINCA-associated NLRP3 mutants exhibit cryo-sensitivity, but this feature is masked by its potent ability to form aggregates. Indeed, other CINCA-associated D303N and Y570C mutants exhibited a potent ability to form aggregates even at 37ºC (Figure 1I). We have discussed this issue in the revised manuscript (page 11, paragraph 1).

Do foci number correlate well with IL-1B release?

We are sorry for our confusing data. Because the number of foci was analyzed in the cells lacking ASC, these cells are defective in mutated NLRP3-mediated IL-1β release. We consider that further refined experimental designs are needed to evaluate the capability of IL-1β release in each NLRP3 mutant.

It might be useful to see if the CINCA mutation has some K and NEK7 dependence.

Thank you for your suggestion. We have moved the NEK7 data to main figure (Figure 4A and B). In addition, we have investigated the NEK7-dependency in the CINCA mutant and found that D303N mutant also induced IL-1β release in the absence of NEK7 (Figure 4—figure supplement 1D and E). We have added these data and mentioned these results in the revised manuscript (page 7, paragraph 1).

Are results the same for CINCA mutations.

Thank you for your helpful comment. According to the suggestion, we investigated ca^2+^ influx induced by CAPS-associated D303N mutant. Similar to the FCAS mutant, induction of the D303N mutant induced ca^2+^ influx (Figure 6—figure supplement 1A and B).

6D Interesting that casp 1 inhibition affects ASC solubility.

Sorry for the lack of explanation. We investigated the effect of pharmacological inhibition of caspase-1 on ca^2+^ influx induced by NLRP3-L353P mutant using fluo-8. The number of fluo-8-positive cells was decreased under the caspase-1 inhibition. In the revised manuscript, we further validated the involvement of caspase-1 using *CASP1*-deficient cells (Figure 7G).

MCC950 data could be moved from Supplemental to main.

According to the suggestion, we have moved the data on MCC950 to main figure. (Figure 8I and J).

Are results the same for CINCA mutations.

Thank you for your helpful comment. According to the suggestion, we have added data on CINCA-associated D303N mutant (Figure 7—figure supplement 1C). The pharmacological inhibition of caspase-1 prevented IL-1β release induced by D303N mutant. However, while pannexin 1 inhibition prevented IL-1β release induced by L353P mutant, the pannexin 1 inhibition failed to prevent IL-1β release induced by D303N mutant (Figure 8A–H and Figure 8—figure supplement 1G). These results suggest that CINCA-associated mutant exhibits constitutive active properties. Thus, feed-forward regulation mediated by ca^2+^ influx promotes inflammasome assembly only in FCAS-associated NLRP3 mutant. We have discussed this issue in the revised manuscript (page 12, paragraph 2).